

**Aerosol hygroscopicity over the South-East Atlantic Ocean during the**
**biomass burning season: Part I – From the perspective of scattering**
**enhancement**
**Lu Zhang[1,2], Michal Segal-Rozenhaimer[1,3,4*], Haochi Che[2*], Caroline Dang[3,4], Junying**
**Sun[5], Ye Kuang[6,7], Paola Formenti[8], Steven G. Howell[9]**
[1]Department of Geophysics, Porter School of the Environment and Earth Sciences, Tel Aviv
University, Tel Aviv, Israel
[2]Department of Geosciences, University of Oslo, Oslo, Norway
[3]Bay Area Environmental Research Institute, Moffett Field, California, USA
[4]NASA Ames Research Center, Moffett Field, California, USA
[5]State Key Laboratory of Severe Weather & Key Laboratory of Atmospheric Chemistry,
Chinese Academy of Meteorological Sciences, Beijing, China
[6]Institute for Environmental and Climate Research, Jinan University, Guangzhou, China
[7]Guangdong-Hongkong-Macau Joint Laboratory of Collaborative Innovation for
Environmental Quality, Guangzhou, China
[8]Université Paris Cité and Univ Paris Est Creteil, CNRS, LISA, Paris, France
[9]Department of Oceanography, University of Hawai'i at Mānoa, Honolulu, USA
Correspondence: Michal Segal-Rozenhaimer (msegalro@tauex.tau.ac.il) and Haochi Che
(haochi.che@geo.uio.no)



**Abstract**
Aerosol hygroscopicity plays a vital role in aerosol radiative forcing. One key parameter
describing hygroscopicity is the scattering enhancement factor, $f$(RH), defined as the ratio of
the scattering coefficient at humidified relative humidity (RH) to its dry value. Here, we utilize
the $f$(80%) from ORACLES 2016 and 2018 airborne measurements to investigate the
hygroscopicity of aerosols, its vertical distribution, its relationship with chemical composition,
and its sensitivity to organic aerosol (OA) hygroscopicity over the South-East Atlantic (SEA)
Ocean during the biomass burning (BB) season.
We found that aerosol hygroscopicity remains steady above 2 km, with a mean $f$(80%) of
1.40±0.17. Below 2 km, aerosol hygroscopicity increases with decreasing altitude, with a mean
$f$(80%) of 1.51±0.22, consistent with higher values of BB hygroscopicity found in the literature.
The hygroscopicity parameter of OA ($\kappa_{OA}$) is retrieved from the Mie model with a mean value
of 0.11±0.08, which is in the middle to upper range compared to literature. Higher OA
hygroscopicity is related to aerosols that are more aged, oxidized, and present at lower altitudes.
The enhanced BBA hygroscopicity at lower altitudes is mainly due to a lower OA fraction,
increased sulphate fraction, and greater $\kappa_{OA}$ at lower altitudes.
We propose a parameterization that quantifies $f$(RH) with chemical composition and $\kappa_{OA}$ based
on Mie simulation of internally mixed OA-$(NH_4)_2SO_4$-BC mixture. The good agreement
between the predictions and the ORACLES measurements implies that the aerosols in the SEA
during the BB season can be largely represented by the OA-$(NH_4)_2SO_4$-BC internal mixture
with respect to the $f$(RH) prediction. The sensitivity of $f$(RH) to $\kappa_{OA}$ indicates that applying a
constant $\kappa_{OA}$ is only suitable when the OA fraction is low and $\kappa_{OA}$ shows limited variation.
However, in situations deviating these two criteria, $\kappa_{OA}$ can notably impact scattering



coefficients and aerosol radiative effect; therefore, accounting for $\kappa_{OA}$ variability is
recommended.
**Keywords:** hygroscopicity, biomass burning aerosol, chemical composition, $\kappa_{OA}$, Atlantic,
airborne measurements, parameterization



## 1 Introduction

Aerosol hygroscopicity is an important physicochemical property of atmospheric aerosols, representing the extent to which particles take up water when exposed to a certain relative humidity (RH) (Covert et al., 1972). Key parameters describing aerosol hygroscopicity include the scattering enhancement factor, $f$(RH), which represents the enhancement of the aerosol light-scattering coefficient as a function of RH (Carrico et al., 2003), and $\kappa$, the hygroscopicity parameter, whose value is defined by its effect on the water activity of the solution (Petters and Kreidenweis, 2007). Water uptake will increase the size and the mass of hygroscopic aerosols, alter their refractive index, enhance the scattering ability, and ultimately influence the single scattering albedo and aerosol radiative forcing (Cotterell et al., 2017; Titos et al., 2021; Zieger et al., 2013). Furthermore, hygroscopicity affects aerosols' ability to act as CCN (cloud condensation nuclei) and ice nuclei, and further influences cloud properties and precipitation (Cai et al., 2021; Che et al., 2017; Ervens et al., 2007). Model results show that even a modest change in $\kappa_{OA}$ ($\kappa$ of organic aerosols) can lead to significant changes in CCN, droplet number concentration, and aerosol radiative effects (Liu and Wang, 2010; Rastak et al., 2017). The treatment of aerosol hygroscopicity is one of the key factors contributing to discrepancies between model simulations and observations and among model estimates (Burgos et al., 2020; Haywood et al., 2008; Reddington et al., 2019).

Africa emits ~ 1/3 of the Earth's annual BB emissions (van der Werf et al., 2010), and its burned areas are increasing every year (Andela et al., 2017). Every Austral spring (July to October), the BB aerosols (BBAs) from African fires are transported westward through the free troposphere (FT) over the persistent stratocumulus cloud deck in the South-East Atlantic (SEA), and eventually subside into the marine boundary layer (MBL) (Redemann et al., 2021). BBAs undergo atmospheric processing during transport, altering their chemical composition, oxidation extent, particle polarity, molecular weight, volatility, and solubility (Rastak et al.,



2017), making the hygroscopicity highly variable. Laboratory studies show that minutes-old
BBAs are more hygroscopic than hour-old BBAs (Day et al., 2006), while the hygroscopicity
of BBAs transported for more than several days in the SEA region remains an area of
investigation. Furthermore, these BBAs mix with pristine aerosols and are subject to marine
influences from the SEA, resulting in a distinct vertical variation of aerosol hygroscopicity.

The hygroscopicity of organic aerosol (OA), the dominant component of aerosols in

most cases, is poorly characterized due to its chemical complexity (Kuang et al., 2020; Mei et
al., 2013). Values of $\kappa_{OA}$ can range from 0 for hydrophobic freshly emitted organics to >1.0 for
very hygroscopic amino acids (Kuang et al., 2020; Petters et al., 2009; Zhang et al., 2007).
BBOA is usually regarded as hydrophobic, while the mass fraction of aged BBOA shows a
positive correlation with $\kappa_{OA}$ (Cerully et al., 2015; Kuang et al., 2021). Several studies have
found a linear correlation between OA hygroscopicity and its oxidation level, commonly
characterized by the oxygen-to-carbon (O/C) ratio or the fraction of total organic mass spectral
signal at m/z 44 ($f_{44}$) (Lambe et al., 2011; Mei et al., 2013). However, this linear relationship
is not always established, especially for secondary OA with a lower O/C ratio under sub-
saturated conditions, for which solubility may play a more important role. In addition, studies
show molecular weight, surface tension, and liquid-liquid phase separation are also related to
the water affinity of OA (Liu et al., 2018; Rastak et al., 2017; Wang et al., 2019), all
contributing to the complexity of OA hygroscopicity.

The ORACLES (ObseRvations of Aerosols above CLouds and their intEractionS)

campaign (Redemann et al., 2021) provides a comprehensive observation of aerosols above the
SEA Ocean with 4-12 days of transport from Africa fires, making it a valuable opportunity to
investigate the hygroscopicity of aged BBA and their OA.  In this paper, we first characterize
the aerosol hygroscopicity and its vertical distribution over the SEA during the BB season, then
propose a parameterization relating aerosol hygroscopicity with chemical composition and $k_{OA}$,



and evaluate the sensitivity of aerosol hygroscopicity to $k_{OA}$. Results are expected to provide a
reference to the treatment of aerosol hygroscopicity in climate models and satellite retrievals,
and to contribute to aerosol-cloud-interactions and radiative assessments in this climatically
important SEA region.
**2 Methods**
2.1 Aircraft Instrumentation and Data Analysis

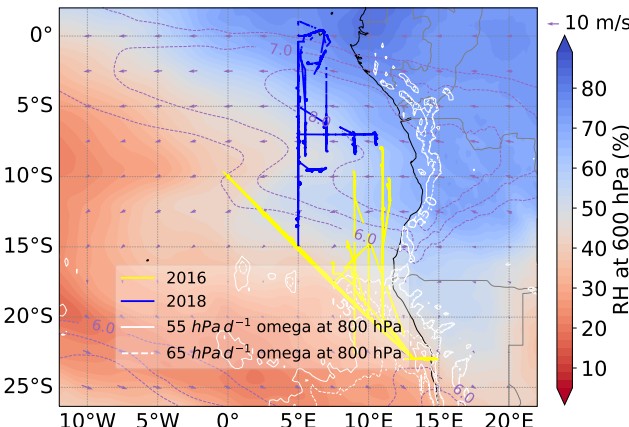


Figure 1. Flight tracks in 2016 and 2018 ORACLES campaigns. Map of October mean of
ERA5 600 hPa RH overlaid by the 600 hPa zonal wind (purple contours; 6, 7, and 8 m s$^{-1}$),
600 hPa horizontal wind vector (purple arrows; m s$^{-1}$), and ORACLES flight tracks in 2016
(yellow) and 2018 (blue), respectively. White contours are the 2016 September mean vertical
velocity, omega, at 800 hPa. Solid and dashed lines represent the subsidence of 55 and 65
hectopascals per day (hPa d$^{-1}$).

We analyzed airborne, in situ data measured over the SEA region from the ORACLES

campaign performed in September 2016 and October 2018 (Redemann et al., 2021). The flight
tracks are shown in Fig. 1. All instruments were deployed on the NASA P-3 aircraft. Two
Radiance Research (RR) M903 integrating nephelometers (Nephs) were operating in parallel,



one (referred to as the 'reference Neph') under relatively dry conditions and the other (known
as the 'humidified Neph') maintained at ~80 % RH. Temperature errors are about 0.5℃ and
RH errors are roughly 3 %. Measurements were reported at 1 Hz. Particles entering the
reference Neph were heated to the aircraft cabin temperature, which significantly reduced their
RH in the Neph. For the calculation of $f$(RH), data with reference Neph RH > 30 % were
excluded. Calibrations were performed in the field with refrigerant R-134A (1,1,1,2-
tetrafluoroethane). All scattering coefficients and scattering enhancement factors are reported
at 540 nm wavelength.

The non-refractory submicron aerosol composition was provided by a High-Resolution

Time-of-Flight Aerosol Mass Spectrometer (HR-ToF-AMS, Aerodyne Research Inc.) (Che et
al., 2022a). The fragment analysis provided $f_{44}$ and $f_{60}$, representing the fractions of the OA
mass spectrum signals at m/z=44 (mainly $CO_2^+$) and m/z=60 (mainly $C_2H_4O_2^+$), respectively,
in the total OA mass. The mass concentration of refractory BC was provided by a single particle
soot photometer (SP2, Droplet Measurement Technology).

The dry particle number size distribution (PNSD) with volume equivalent diameter

ranging from ~90 nm to 10 μm was obtained by combining measurements from an ultra-high-
sensitivity aerosol spectrometer (UHSAS) and an aerodynamic particle sizer (APS). The
UHSAS undersized particles and the data were corrected using Howell et al. (2021). The
aerodynamic diameter of APS was converted to the volume equivalent diameter according to
DeCarlo et al. (2004). Particles were assumed to be spherical (shape factor = 1) with a density
of 1.5 g cm⁻³. The aerosol/plume age was modelled with a two-week forecast using the Weather
Research and Aerosol Aware Microphysics (WRF-AAM) model (Thompson and Eidhammer,
2014). Carbon monoxide was tagged as tracer at the fire source, identified by a burned area
product from the moderate resolution imaging spectrometer with a 500 m spatial resolution.



All measurements are averaged to 15 s and adjusted to STP conditions at 273.15 K and
1013 hPa. Data with scattering coefficient < 10 Mm$^{-1}$ are not included. $f$(RH) with RH>30 %
for the reference Neph or RH<76 % for humidified Neph are also excluded. The final
measurements used in this study have an average RH of 79±0.5 % for the humidified Neph and
RH<30 % for the reference Neph. To ensure the influence of BB emissions, only data with
$f_{60}$>0.003 are considered (Cubison et al., 2011). This study analyzes measurements from 21
flights totaling approximately 134 flight hours.
2.2 Calculation of $f$(RH) and $\gamma$ parameterization
The aerosol scattering enhancement factor, $f$(RH), is calculated as:

$$f(RH) = \frac{\sigma_{sp}(RH)}{\sigma_{sp}(RH_{ref})} \tag{1}$$

where $\sigma_{sp}$(RH) and $\sigma_{sp}$(RH$_{ref}$) represent the scattering coefficients at humidified and reference
RHs, respectively. Note the $f$(RH) only include those with reference RHs equal to or smaller
than 30 % to facilitate comparison with previous studies. For simplicity, we denote the $f$(RH)
at the RH of humidified Neph as $f$(80%), despite the small variation of the RH in humidified
Neph. The $f$(RH) is usually fitted to a $\gamma$ parameterization to apply to a more extensive RH range
(Sheridan et al., 2002; Titos et al., 2016):

$$f(RH) = (\frac{1 - RH/100}{1 - RH_{ref}/100})^{-\gamma} \tag{2}$$

In our case, the $\gamma$ was calculated with the RH and RH$_{ref}$ using Eq. 2 since the $f$(RH) was only
measured at a fixed RH.
2.3 $\kappa_{f(RH)}$ retrieval and $\kappa_{OA}$ calculation
The aerosol hygroscopicity parameter $\kappa$ can be retrieved from $f$(RH), usually denoted
as $\kappa_{f(RH)}$ (Chen et al., 2014). It can be regarded as the scattering coefficient weighted average $\kappa$
(Kuang et al., 2021). The dry scattering coefficient can be computed using Mie theory. The



Python package PyMieScatt (Sumlin et al., 2018), an implementation of the Mie theory (Mie,
1908), was applied in this study. Inputs of the Mie model include particle refractive index and
PNSD. Particles beyond PM$_1$ (particulate matter with an aerodynamic diameter less than 1 μm)
are not included in this calculation, which can be supported by their small contribution to the
total volume (average < 3 %). By combining Mie model with the $\kappa$-Köhler theory, we can then
calculate the scattering coefficients under humidified RH conditions. Subsequently, $f$(RH) and
$\gamma$ can be obtained using Eq. 1 and 2. In the calculation, a volume mixing rule was used to
calculate the refractive index under both dry and humidified conditions. The volume of
inorganic salts was converted from those of $SO_4^{2-}$, $NO_3^-$, and $NH_4^+$ from AMS following a
modified ion-pairing scheme (Gysel et al., 2007; Zhang et al., 2022). The hygroscopic
parameter $\kappa$ and density can be found in Table S1. We iteratively adjust $\kappa_{f(RH)}$ to minimize the
difference between the calculated and measured $f$(RH). Detailed descriptions of the retrieval
procedure of $\kappa_{f(RH)}$ can be found in Chen et al. (2014).
According to Petters and Kreidenweis (Petters and Kreidenweis, 2007), the overall $\kappa_{chem}$
can also be calculated from various chemical compositions following the ZSR (Zdanovskii-
Stokes-Robinso) mixing rule. Kuang et al. (2020b) thoroughly outlined in Section 3.3 that the
$\kappa_{f(RH)}$ can accurately represent the $\kappa_{chem}$ of PM$_1$. Therefore, the hygroscopicity parameter of OA,
$\kappa_{OA}$, can be calculated as:

$$\kappa_{OA} = \frac{\kappa_{f(RH)} - (\sum_{inorg} \kappa_i \varepsilon_i + \kappa_{BC} \varepsilon_{BC} + \kappa_X \varepsilon_X)}{\varepsilon_{OA}}, \tag{3}$$

where *inorg* represents inorganic salts. ε represents the volume fraction of each component.
**3 Results and discussion**
3.1 Overview of chemical compositions in 2016 and 2018 ORACLES

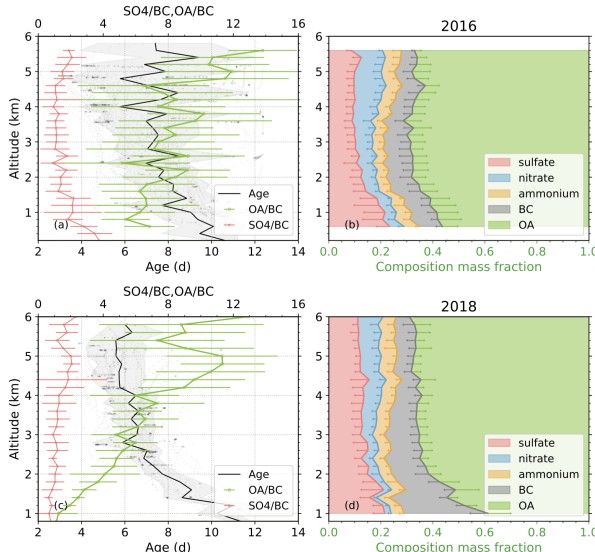


Figure 2. The vertical distribution of plume age and chemical composition. (a, c) Variation of

plume age (black), OA/BC, and SO4/BC with altitude in 2016 (upper) and 2018 (lower)

ORACLES campaigns, respectively. Grey dots show the distribution of plume age with the

altitude. (b, d) The average vertical distribution of the mass ratio of chemical compositions

from AMS and SP2 in every 200 m in 2016 and 2018 ORACLES campaigns, respectively. The

lines are the mean value in every 200 m bin. Errorbars and grey shading represent the standard

deviation in every 200 m bin.

Flights in 2016 ORACLES (Fig. 1, yellow lines) are in the region of 8-24° S and 0-15°

E, traversing both the southern African Easterly Jet (AEJ-S) region and the continent

anticyclone. As a result, aerosols around 600 -700 hPa in 2016 ORACLES include both less

aged (<4 d) particles coming directly from the continent and highly aged (>10 d) particles

transported from the west/north, resulting in a larger variation of plume age in each level as

shown in Fig. 2a. At lower altitudes, aerosols are less aged than those in the 2018 campaign

due to the subsidence near the Namibian coast (Fig. 1a). During the 2016 campaign, the cloud

top is generally below 1.5 km. The 2018 ORACLES flights, represented by blue lines in Fig.



1, are primarily situated within the 0-15° S and 5-10° E coordinates. The cloud top in this
region is a bit lower than in 2016 campaign, centering around 1 km. This area generally
coincides with the region influenced by the southern African Easterly Jet (AEJ-S). BB aerosols
are lifted up to the free troposphere, transported westward by AEJ-S and then subside into the
marine boundary layer, rendering the distinct vertical age pattern that increases with the
decreasing altitude (Fig. 2c). Correspondingly, aerosols in the SEA region during BB season
exhibit distinct vertical distribution of chemical composition. From Fig. 2b and 2d, the vertical
profiles of chemical composition fractions are generally consistent during 2016 and 2018
ORACLES campaigns. In this section, we focused on the variation of OA and sulphate, two
components that dominate aerosol hygroscopicity in the SEA.

OA constitutes the largest fraction of aerosol mass in ORACLES, approximately 60 %.

The OA mass fraction in both years shows little variation above 2 km; below this altitude, OA
mass fraction decreases with decreasing altitude, in contrast to the trend of the sulphate mass
fraction. The OA/BC ratio, representing the OA mass concentration normalised by that of BC
to remove the dilution effect during transport, differs in 2016 and 2018. While 2018 data shows
a clear decrease in OA/BC with decreasing altitude, the decrease was less pronounced in 2016,
showing considerable variation at identical altitudes. Dobracki et al. (2022) used RH as an
indicator to investigate the importance of thermodynamic partitioning in OA/BC changes
during the 2016 ORACLES campaign, concluding that it accounts for no more than 10 % of
the changes. The dominant factor is believed to be the oxidation of OA through fragmentation.
A similar result is found in this study using temperature as an indicator, as shown in Fig. 3a.
Please note Fig. 3 only considers OA above 1.4 km and temperature > 0 °C to minimize the
marine influence and to exclude possible ice nucleation. The OA/BC ratio in the 2016
ORACLES campaign did not show a clear decrease with increasing temperature, as $NO_3$/BC
did, which is a result of thermodynamic repartition to the gas phase. However, in the 2018



ORACLES campaign, we did notice a significant decrease of OA/BC with increasing
temperature (Fig. 3b). The OA/BC decreased ~70 % from 9.7±3.1 for temperature 0-4 °C to
2.9±0.9 for temperature > 20 °C, only slightly lower than the decrease of $NO_3$/BC, ~85 %. Yet,
we cannot simply attribute the OA/BC changes to thermodynamic repartition while
disregarding the effect of ageing or OA oxidation. In 2018, temperature and plume age are
closely correlated (Pearson correlation coefficient of 0.51), and the decrease in OA/BC is
accompanied by ageing (Pearson correlation coefficient of 0.57), as shown in Fig. 2a and b.
We utilized the oxidation state to differentiate between the effects of thermodynamic
repartition and OA oxidation. Figure 4 shows the Van Krevelen diagrams (H/C vs. O/C, Ng et
al., 2011) for aerosols under temperatures > 20 °C and 0-4 °C. The estimated carbon oxidation
state ($OS_C$), defined as $OS_C$=2O/C−H/C, can also indicate different OA volatility regimes, with
OSc of -2.0− -1.5 for HOA (hydrocarbon-like OA), -1.75− -0.75 for BBOA (biomass burning
OA), -1.0−0.0 for SV-OOA (semi-volatile oxidized OA), and 0.0−1.0 for LV-OOA (low
volatility oxidized OA) (Donahue et al., 2012; Kroll et al., 2011). If thermodynamic repartition
plays a more crucial role, the OA remaining under higher temperature would be less volatile
due to evaporation of more volatile OA. Notably, we found the opposite. From Fig. 4, aerosols
under temperature > 20 °C (lower altitudes) are generally more volatile than those at
temperature 0-4 °C (higher altitudes). This indicates that thermodynamic repartition is not a
dominant factor in OA/BC changes, and that the OA oxidation through fragmentation is more
important in OA/BC changes in 2018, consistent with the 2016 campaign as well as results in
Dobracki et al. (2022). This is also in line with the findings of Dang et al. (2022) which found
less organics in aerosols collected on filters associated with more aged plumes and more
rounded and viscous organics on filters sampled from less aged plumes. For OA below 1.4 km,
aqueous phase reactions and cloud scavenging might also contribute to the loss of OA during
the entrainment and within the MBL (Che et al., 2021; Wu et al., 2020).




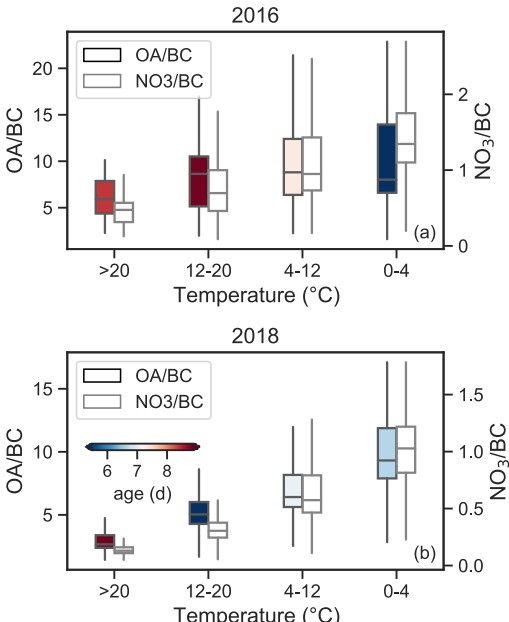


Figure 3. OA/BC (black outline) and NO$_3$/BC (grey outline) mass ratios as a function of

ambient temperature in 2016 (a) and 2018 (b) ORACLES campaign, for altitude > 1.4 km and

temperature > 0 °C. The boxes represent the 10[th] percentile, 25[th] percentile, median, 75[th]

percentile, and 90[th] percentile.

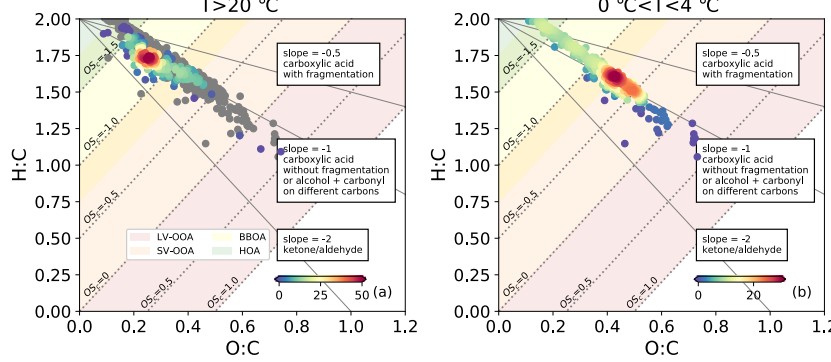


Figure 4. Van Krevelen diagram (H/C vs. O/C) for aerosols with temperature higher than 20 °C

(a) and with temperature lower than 4 °C (b). The color scale indicates the density of the data





in each plot. The grey dots in (a) are the Van Krevelen diagram of aerosols with temperature
lower than 4 °C, the same as (b).

The variation of sulphate mass fraction remains largely constant above 2 km, and below

2 km, increases with decreasing altitudes. The higher sulphate fraction at lower altitudes is
consistent with the observations from CLARIFY-2017 (CLoud-Aerosol-Radiation Interaction
and Forcing for Year 2017) campaign (Wu et al., 2020), which was conducted downwind of
ORACLES in the SEA ocean. This higher sulphate fraction at lower altitudes results from the
increase of $SO_4$/BC and decrease of OA/BC. $SO_4$/BC ratio generally remains constant above
800 m in both years' campaign. However, for 2016 ORACLES campaign, where there are
samples below 800 m, the ratio shows an increase with decreasing altitude. This increase could
indicate a sulphate contribution from the ocean, either in the form of sea-salt sulphate or
through dimethylsulfide (DMS) emitted by marine phytoplankton. The latter can contribute to
non-sea-salt sulphate by oxidizing to $SO_2$ and further to sulphate (Mayer et al., 2020; Alexander
et al., 2005). Notably, part of the 2016 flight region, especially the SEA offshore of Namibia,
is known as an upwelling region with high DMS emissions (Andreae et al., 1995). Klopper et
al. (2020) have attributed 57 % of sulphate to sea salt and 43 % to non-sea-salt sulphate along
the Namibian coast. These findings align with model simulations showing that DMS is the third
largest CCN source in the SEA up to 2 km (Che et al., 2022b).

Furthermore, BC mass constitutes approximately 10 % of the $PM_1$ mass fraction,

indicating the large influence of BB in this region. The nitrate mass fraction increases with
increasing altitude in all layers, which is consistent with the findings of CLARIFY, and can be
explained by the shift of gas-particle partitioning of the $HNO_3$-$NH_3$-$NH_4NO_3$ system towards
the aerosol phase at the lower temperatures found at higher altitudes (Wu et al., 2020). The
mass fraction of ammonium stays stable with height, approximately 5 %.
3.2 Aerosol hygroscopicity in SEA in 2016 and 2018 ORACLES



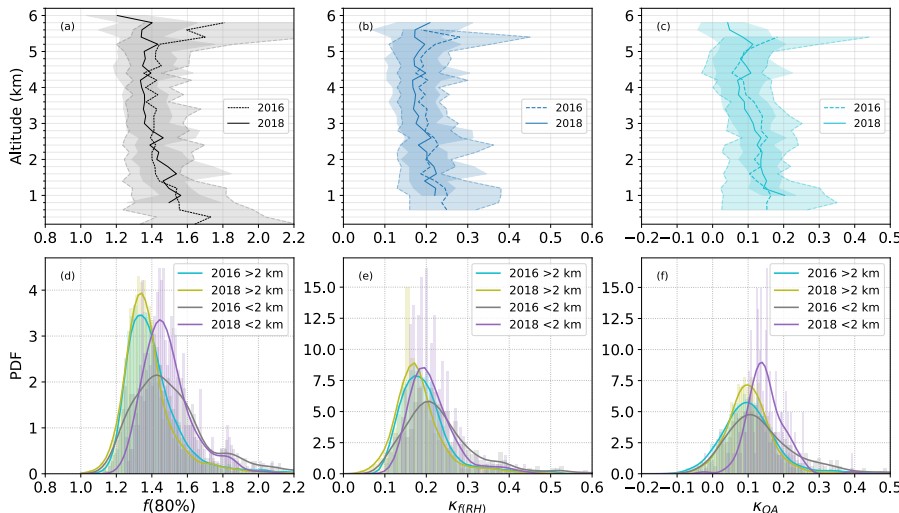


Figure 5. Vertical profiles and PDF of $f(80\%)$ (a, d), $\kappa_{f(RH)}$ (b, e), and $\kappa_{OA}$ (c, f) for aerosols in

the 2016 (dotted line) and 2018 (solid line) ORACLES campaign. Dashed lines in a, b, and c

represent the 10th percentile, mean, and 90th percentile, respectively.

In general, the aerosol hygroscopicity stays stable above 2 km in both years' campaigns;

while below 2 km, aerosols become more hygroscopic at lower altitudes (Fig. 5). This is

consistent with the vertical variation of sulphate and OA mass fraction, i.e. more sulphate and

less OA at lower altitudes. The probability density function (PDF) distributions of $f(80\%)$ and

$\kappa_{f(RH)}$ are similar in the 2016 and 2018 campaigns, with larger variations and higher values of

the aerosol hygroscopicity PDF under 2 km (Fig. 5d and 5e). For $f(80\%)$ below 2 km, a primary

mode with a diameter around 1.45 is evident, but there is also a second mode with a diameter

around 1.81 for aerosols in both years. While the second mode is subtle, it can be identified in

the PDF of $\kappa_{f(RH)}$ (Fig. 5e). This suggests the presence of highly hygroscopic substances and

could indicate marine influence, as most aerosols below 2 km are within the MBL. For aerosols

above 2 km, the mean and standard deviation of $f(80\%)$ and $\kappa_{f(RH)}$ are 1.40±0.17 and 0.19±0.07

(Fig. 5 and Table S2), respectively, belonging to less hygroscopic particles (Liu et al., 2011).



These values are generally lower than those for marine aerosols (Zieger et al., 2010; Carrico et
al., 2003), higher than dust and polluted dust particles (Bukowiecki et al., 2016; Zhang et al.,
2015a), and consistent with the median level of the hygroscopicity for smoke-dominated
aerosols found in the literature. They are comparable to the $f$(80%) of 1.37 for smoke from
lightly-wooded savanna fires in Australia ($D_P$<3 μm) (Gras et al., 1999), and the $f$(82%) of 1.40
for BBAs from forest fires in northeast US (Wang et al., 2007); while slightly higher than the
$f$(80%) of BBAs in Brazil (SCAR-B, $D_P$<4 μm) (Kotchenruther and Hobbs, 1998). For aerosols
below 2 km, they belong to more hygroscopic particles (Liu et al., 2011). The mean and
standard deviation of $f$(80%) and $\kappa_{f(RH)}$ are 1.51±0.22 and 0.23±0.08, respectively, which
belong to the upper ranges of BBA hygroscopicity in the literature. These values are
comparable to the $f$(80%) of 1.42±0.05 for smoke collected between 10 and 50 min of emission
in Africa (SAFARI, $D_P$<5 μm) (Magi and Hobbs, 2003), the $f$(80%) of 1.43±0.12 in a
background station in the Yangtze River Delta of China (Zhang et al., 2015a), and the $\kappa_{f(RH)}$ of
0.22±0.04 in a rural site in southern China (Kuang et al., 2021); while lower than the $f$(85%)
of 1.60±0.20 for BBAs in East Asia during ACE-Asia (Asian Pacific regional aerosol
characterization experiment) (Kim et al., 2006), the $f$(85%) of 1.58±0.21 for agricultural
burning in INDOEX (Indian Ocean Experiment) (Sheridan et al., 2002), the $f$(80%) of
1.66±0.08 for fresh smoke (within 10 min from emission) in Africa (Magi and Hobbs, 2003).
Comparing to the $\kappa$ obtained from CCN measurements at a similar location in August 2017
ORACLES (Kacarab et al., 2020), our results are ~ 30 % lower. This difference is expected
because $\kappa$ values obtained under supersaturated conditions are typically larger than those from
sub-saturated conditions (Petters and Kreidenweis, 2007). This highlights the significance of
using the appropriate $\kappa$ for sub-saturated and supersaturated investigations, such as when
examining aerosol liquid water content and cloud condensation nuclei activation (Rastak et al.,
2017; Petters and Kreidenweis, 2007).





The mean $\kappa_{OA}$ (±1 standard deviation) is 0.11±0.08, with the 25th and 75th percentiles
of 0.06 and 0.16. From the vertical profiles, more hygroscopic OA are generally more aged,
highly oxidized, and usually located at lower altitudes (Fig. 2 and 5). In addition, we observed
a slight increase in $\kappa_{OA}$ with volatility in 2016, with a Pearson correlation coefficient of -0.35
between $\kappa_{OA}$ and OSc, contrasting the conventional understanding that the most volatile
compounds have the least hygroscopicity. This trend has been observed, albeit rarely, in field
and laboratory studies (e.g. Cerully et al., 2015; AsaAwuku et al., 2009). It may be related to
fragmentation during OA oxidation, where the highly aged and low volatile OA may dissociate
into more volatile fragments that are still highly functionalized and hygroscopic. However, in
general, no clear correlation has been found between $\kappa_{OA}$ with altitude or oxidation level.
We noted a portion of highly aged aerosols (> 10 d) in 2016 having high OA/BC (> 12,
corresponding OA mass fraction > 50 %), in contrast to the general trend that more aged
aerosols correspond to smaller OA/BC (Fig. 2). About 95 % of these aerosols are above 3 km
and have a slightly lower $f_{44}$ than the campaign average (Fig. S1a). Approximately 60 % belong
to LV-OOA with OSc > 0 and 40 % are SV-OOA (Fig. S1b). As shown in Fig. S1c, the $\kappa_{OA}$
values are smaller for these aerosols compared to the whole 2016 campaign, which is consistent
with previous studies that $\kappa_{OA}$ is lower for less oxidized OA (Kuang et al., 2020; Rastak et al.,
2017; Mei et al., 2013); though we do not observe such correlation for the entire campaign. We
hypothesize that thermodynamical repartitioning has played a role, i.e. less-oxidized materials
condensed onto pre-existing OA under low temperature at high altitudes, resulting in smaller
$f_{44}$ values and contributing to SV-OOA. These less-oxidized materials are generally less
functionalized and less hygroscopic, which would lead to a lower $\kappa_{OA}$.
3.3 Relationship with chemical composition and $\kappa_{OA}$
3.3.1 Comparison with various campaigns



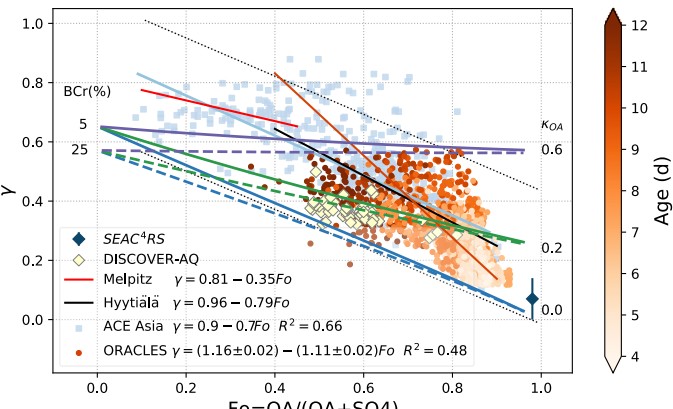


Figure 6. γ versus Fo in various campaigns and for internally mixed OA-(NH₄)₂SO₄-BC

mixtures. Fo represents the ratio of mass concentrations of OA to OA and $SO_4^{2-}$. Solid lines in

light blue and brown represent the linear fits for ACE-Asia and ORACLES, respectively.

Dotted grey lines show the 95% prediction bands for the ACE-Asia data, in light blue

rectangles, taken from Quinn et al. (2005). Colorbar represents the plume age (days) in

ORACLES. Data for SEAC⁴RS is shown by blue diamond, taken from Shingler et al. (2016).

DISCOVER-AQ data is shown by yellow diamonds, taken from NASA Langley Research

Center Atmospheric Science Data Center (Atmospheric Science Data Center, 2015). Fitting

lines for two European sites Melpitz (solid red line) and Hyytiälä (solid black line) are from

Zieger et al. (2015). Blue, green, and purple lines represent results for internally mixed OA-

(NH₄)₂SO₄-BC mixtures with 1) a range of BC mass fraction (BCr, solid for 5% and dashed

for 25%) and 2) OA with $\kappa_{OA}$ of 0 (blue), 0.2 (green), and 0.6 (purple) from Mie calculations

assuming a lognormal size distribution with a geometric mean diameter D_gn of 150 nm and a

standard deviation σ_sg of 1.5.

Quinn et al. (2005) proposed a parameterization quantifying the relationship between γ

and Fo, the ratio of mass concentrations of OA to OA and $SO_4^{2-}$, based on measurements in

ACE-Asia. We applied the parameterization to ORACLES measurements and as shown in Fig.



6, our data are well within the 95% prediction confidence intervals. We further investigated the
γ-Fo dependence of BBAs from DISCOVER-AQ and SEAC[4]RS (Shingler et al., 2016) and
continental aerosols from the central European station Melpitz and a boreal site Hyytiälä in
Finland (Zieger et al., 2014, 2015), all showed good overlap with those from ACE-Asia and
ORACLES. The linear regression for ORACLES, $\gamma = (1.16\pm0.02) - (1.11\pm0.02)\cdot Fo$, retrieved
from an orthogonal fit by taking the standard deviation as the input for uncertainty calculation,
is very similar to those in Hyytiälä and ACE-Asia, though the slope is slightly lower.

We explored the γ-Fo relationship with the Mie model and found that the relationship

observed can be largely explained by aerosol chemical composition and OA hygroscopicity.
The γ values were calculated with the scattering coefficients simulated at both dry conditions
and 80 % RH were performed with Mie model for internally mixed OA-$(NH_4)_2SO_4$-BC
mixtures with assumed BC mass ratio (BCr, 5 % and 25 %), and $\kappa_{OA}$ values (0-0.6), which
encompass the ranges observed in ORACLES (refer to Sect. 3.2 for $\kappa_{OA}$ values). The PNSD
was calculated following the lognormally distribution, with the geometric mean diameter ($D_{gn}$)
and standard deviation ($\sigma_{sg}$) set to $D_{gn}$=150 nm and $\sigma_{sg} = 1.6$, respectively. As shown in Fig. 6
(solid and dashed purple, green, and blue lines), simulated curves can capture most of the
observations. Fo and $\kappa_{OA}$ dominant γ, and BC shows a small negative impact. It is noteworthy
that the (negative) slope of the γ-Fo relationship increases with increasing $\kappa_{OA}$ up to $\kappa_{OA}$ values
of 0.6, where γ exhibits little variation with Fo. Therefore, we conclude that the variation of
BBA hygroscopicity with ageing in the SEA is mainly due to changes in chemical composition,
particularly sulphate and OA, as well as the variation of OA hygroscopicity during transport.
The higher BC fraction in aged aerosols compared to less aged ones has slightly decreased the
hygroscopicity of aged aerosols.

3.3.2 Parameterization of γ using Mie simulations of internally mixed OA-$(NH_4)_2SO_4$-

BC mixtures



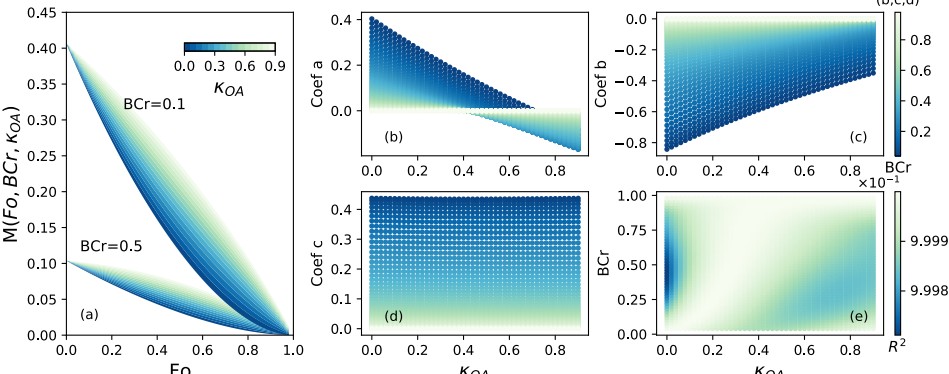

Figure 7. (a) Variations of M(Fo,$\kappa_{OA}$,BCr) with Fo coloured by $\kappa_{OA}$ at BCr of 0.1 and 0.5,

respectively, for internally mixed OA-$(NH_4)_2SO_4$-BC mixtures. M(Fo,$\kappa_{OA}$,BCr) is the product

of γ(Fo,$\kappa_{OA}$=0,BCr) and γ(Fo,$\kappa_{OA}$,BCr) for each $\kappa_{OA}$ value. Fo represents the ratio of the mass

concentration of OA to that of OA and $SO_4^{2-}$. BCr is the mass ratio of BC. (b,c,d) Variation of

coefficients a, b, and c with $\kappa_{OA}$ and BCr. The coefficients a, b, and c are the fitted parameters

of the quadratic regression between M(Fo,$\kappa_{OA}$,BCr) and Fo for each $\kappa_{OA}$ and BCr. (e) The $R^2$

(colorbar) of the M(Fo,$\kappa_{OA}$,BCr) regression with Fo as a function of $\kappa_{OA}$ and BCr.

Mie simulations are performed for internally mixed OA-$(NH_4)_2SO_4$-BC mixtures to

obtain the scattering coefficient of dry and humidified aerosols. We assume PNSD to be a log-

normal distribution with $D_{gn}$=150 nm and $\sigma_{sg}$ = 1.6, as the approximation of the $D_{gn}$ and $\sigma_{sg}$ in

ORACLES 2016 and 2018 campaigns. The RH and $RH_{ref}$ is set as 80 % and 0, respectively.

The γ is then calculated following Eq. 2. The Fo, $\kappa_{OA}$, and BCr are varied from 0 to 1, 0 to 0.9,

and 0 to 1, respectively, all with a span of 0.02. Taking γ(Fo,$\kappa_{OA}$=0,BCr) as the baseline (refer

to solid and dashed blue lines in Fig. 6), we calculated the product M(Fo,$\kappa_{OA}$,BCr) of

γ(Fo,$\kappa_{OA}$=0,BCr) and γ(Fo,$\kappa_{OA}$,BCr) for each $\kappa_{OA}$ and BCr, i.e. M(Fo,$\kappa_{OA}$,BCr)=

γ(Fo,$\kappa_{OA}$=0,BCr)* γ(Fo,$\kappa_{OA}$,BCr), and found that the relationship between M(Fo,$\kappa_{OA}$,BCr) and

Fo can be well fitted into a quadratic (second-order) polynomial function, i.e. M(Fo,$\kappa_{OA}$,BCr)

= $aFo^2$+bFo+c (Fig. 7a). The variation of M(Fo,$\kappa_{OA}$,BCr) with Fo and the $R^2$ of the regression



are shown in Fig. 7a and 7e, respectively. The fitted coefficients a, b, and c, as shown in Fig.
7b, 7c, and 7d, coincidentally fit well as quadratic functions of $\kappa_{OA}$, whose coefficients, in turn,
can be well fitted into a fifth-order polynomial function of BCr. Results are shown in Fig. S3
in the supplement. In sum, the M(Fo,$\kappa_{OA}$,BCr) can be parameterized as:

$$M(F_o, \kappa_{OA}, BCr) = \sum_{\substack{i \leq 2 \\ j \leq 2 \\ k \leq 5}} a_{ijk} BCr^k \kappa_{OA}^j F_o^i$$

(4)

Similarly, γ(Fo,$\kappa_{OA}$=0,BCr) can be well fitted into a quadratic function of Fo with coefficients
that fit well with a fifth-order polynomial function of BCr:

$$\gamma(F_o, \kappa_{OA} = 0, BCr) = \sum_{\substack{i \leq 2 \\ k \leq 5}} a_{ik} BCr^k F_o^i$$

(5)

Equations 4 and 5 in matrix format are referred to Eq. S1 and S2 in the supplement,
respectively. Values of coefficients $a_{ijk}$ and $a_{ik}$ are shown in Table S3. Therefore,
γ(Fo,$\kappa_{OA}$,BCr) can be calculated as the ratio of M(Fo,$\kappa_{OA}$,BCr) to γ(Fo,$\kappa_{OA}$=0,BCr):

(6)

$$\gamma(Fo, \kappa_{OA}, BCr) = M(F_o, \kappa_{OA}, BCr)/\gamma(F_o, \kappa_{OA} = 0, BCr)$$

The $f$(RH) can then be calculated with Eq. 2. We evaluated this parameterization by comparing
the predicted and measured $f$(80%) in ORACLES 2016 and 2018 campaigns. The predicted
$f$(80%) is calculated with Eq. 6 with Fo, $\kappa_{OA}$, and BCr as inputs and Eq. 2 with the dry and
humidified RHs measured in both campaigns. Note the mean BC mass ratio for each year has
been used in the calculation, as little difference has been observed using the temporal BCr and
mean BCr. Good correlation of measured and predicted $f$(80%) has been achieved for both
years' campaign, as shown in Fig. 8a. This indicates that the internally mixed OA-$(NH_4)_2SO_4$-
BC mixture with PNSD ($D_{gn}$=150 nm and $\sigma_{sg}$ = 1.6) is a good approximation of aerosols with
respect to the $f$(RH) prediction in 2016 and 2018 ORACLES campaign. The influence of PNSD
on $f$(RH) is small and discussed in Section S1 in the supplement.



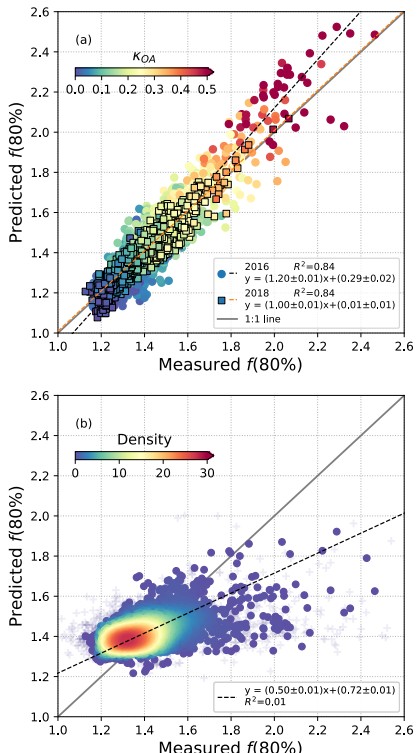


Figure 8. Measured $f(80\%)$ vs predicted $f(80\%)$ using the $\gamma$ parameterization for internally mixed OA-$(NH_4)_2SO_4$-BC mixtures. The $f(80\%)$ in subplot (a) is calculated with $\kappa_{OA}$ values coloured by $\kappa_{OA}$, and in subplot (b) is predicted with the mean $\kappa_{OA}$ values. Black and orange dashed lines in subplot (a) represent the ordinary linear regression for 2016 and 2018, respectively. The black dashed line in subplot (b) represents the ordinary linear regression for the two years. Grey solid line is the 1:1 line.

### 3.3.3 Sensitivity of aerosol scattering enhancement to $\kappa_{OA}$

Due to the chemical complexity of OA, the $\kappa_{OA}$ values of particles are not easily obtained. Various hygroscopicity parameterizations have been proposed in previous studies, most of which are parameterized with chemical composition, e.g. organics or inorganics fraction, and a constant assumed $\kappa_{OA}$ value. Few studies consider the variation of $\kappa_{OA}$ (Zhang



et al., 2015b; Huang et al., 2022). While these parameterizations can represent their
observations well, they may not be suitable for situations with different $\kappa_{OA}$ values. Therefore,
in this section, the influence of $\kappa_{OA}$ on the prediction of $f$(RH) is analyzed. We calculated the
$f$(80%) with the mean $\kappa_{OA}$ in each campaign and the results are shown in Fig. 8b. The use of a
constant $\kappa_{OA}$ average leads to a much smaller variation of the predicted $f$(80%) values, with
most of which concentrated around 1.3-1.4. Predicted $f$(80%) tend to overestimate lower
$f$(80%) values while underestimate higher $f$(80%) values. A slope of 0.50 and a $R^2$ of 0.01
indicates poor prediction in capturing the trend of $f$(80%). This indicates that using Fo, BCr,
and a constant $\kappa_{OA}$ is insufficient for the prediction of $f$(RH), and that the variation of $\kappa_{OA}$ need
to be considered, at least for situations where $\kappa_{OA}$ has a large variation, such as in ORACLES.
To quantitively investigate the sensitivity of $f$(RH) to $\kappa_{OA}$, we calculated the deviation
of $f$(80%) with $\kappa_{OA}$ for the OA-$(NH_4)_2SO_4$-BC mixture. As shown in Fig. 9, we observed that
$\kappa_{OA}$ is positively correlated with $f$(80%). Additionally, the deviation of $f$(80%) is dependent on
the OA fraction (Fo), i.e. a higher OA fraction leads to a larger impact of $\kappa_{OA}$ and consequently
a larger deviation of $f$(80%).
The 25th and 75th percentiles of Fo in 2016 and 2018 ORACLES campaign were 0.74
and 0.86, respectively. These are relatively high values and therefore result in relatively high
spread of $f$(80%). As well, the age of ORACLES OA spans from <4 days to > 10 days, during
which OA oxidation and fragmentation (as discussed in Section 3.2) takes place. These
processes alter the hygroscopicity of OA, causing the OA in ORACLES to contribute to large
variations of $\kappa_{OA}$. These large variations of $\kappa_{OA}$, combined with the relatively high OA fraction
(Fo), makes $f$(RH) highly sensitive to the $\kappa_{OA}$ value. For aerosols with a $\kappa_{OA}$ of 0.4 and a Fo of
0.86, the $f$(80%) can be 80 % higher compared to aerosols with hydrophobic OA, as shown in
Fig. 9. In other words, the aerosol scattering coefficients at 80 % RH are 80 % higher solely
because of the increase of OA hygroscopicity. This high sensitivity also explains the poor



prediction of $f(80\%)$ when using campaign mean $\kappa_{OA}$ values, as shown in Fig. 8b. We further
analyzed the influence of $\kappa_{OA}$ value on $f(80\%)$ for a relatively polluted site on the North China
Plain based on their $\kappa_{OA}$ values (Kuang et al., 2020). Its $\kappa_{OA}$ rises from 0.02 in the morning to
0.16 in the noon, combining its Fo of 0.86, the $f(80\%)$ can increase by 25 % when the $\kappa_{OA}$ is
0.16 compared to it being 0.02. It means that the scattering coefficients at 80 % RH can be
25 % higher at noon compared to the morning solely due to the increase of OA hygroscopicity.
Many studies overlook the variability of $\kappa_{OA}$ and instead use a constant $\kappa_{OA}$ when analyzing
aerosol hygroscopicity or radiative forcing. As illustrated in Fig. 9, this can be reasonable when
the OA fraction is low and $\kappa_{OA}$ exhibits minimal variation; however, in cases where these two
conditions are not met, $\kappa_{OA}$ can significantly influence the scattering coefficients and hence
direct radiative forcing.

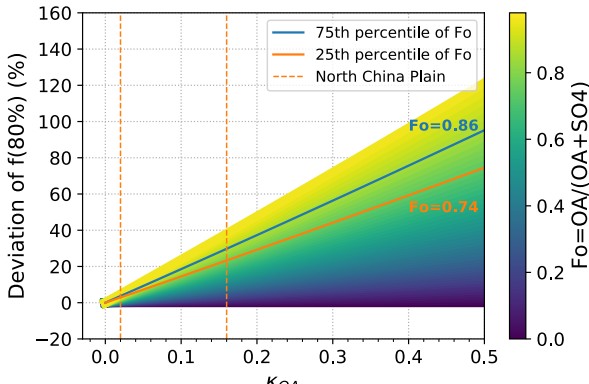


Figure 9. Sensitivity of $f(80\%)$ to $\kappa_{OA}$. The OA to OA + SO4 ratio (Fo) is represented by the
colorbar. The blue and orange lines represent the variation at 75[th] and 25[th] percentile of Fo in
both years' ORACLES campaign, respectively. The pair of dashed orange lines represent the
range of $\kappa_{OA}$ observed at the site on the North China Plain (Kuang et al., 2020).
**4 Conclusion**



The hygroscopicity of aerosols from the perspective of scattering enhancement over the

SEA Ocean during the BB season are investigated using measurements from the 2016 and 2018

ORACLES campaigns. The vertical distribution of aerosol hygroscopicity shows a consistent

pattern in both campaigns, remaining stable above 2 km; below 2 km, aerosols are more

hygroscopic at lower altitudes. Aerosols above 2 km have a mean and standard deviation of

$f(80\%)$ and $\kappa_{f(\mathrm{RH})}$ of 1.40±0.17 and 0.19±0.07, respectively, and are less hygroscopic.

Conversely, aerosols below 2 km are more hygroscopic, and have a mean and standard

deviation of $f(80\%)$ and $\kappa_{f(\mathrm{RH})}$ of 1.51±0.22 and 0.23±0.08, respectively, which are values at

the upper level of BBA hygroscopicity found in the literature. This variation of aerosol

hygroscopicity is consistent with the vertical variation of chemical composition. The OA and

sulphate mass fraction in both years show little variation above 2 km; while below this altitude,

OA decreases with decreasing altitude, while the sulphate mass fraction tends to increase. OA

oxidation through molecular fragmentation is the main mechanism for OA losses in the FT.

While the increase of sulphate in the MBL could indicate marine influence.

We retrieved $\kappa_{\mathrm{OA}}$ using Mie simulations. It shows a large variation, with the mean and

standard deviation being 0.11±0.08 and the 25[th] and 75[th] percentiles of 0.06 and 0.16,

respectively. No clear relationship was found between $\kappa_{\mathrm{OA}}$ and OA oxidation level; while a

slight increase in $\kappa_{\mathrm{OA}}$ with volatility is shown in 2016, which may be related to the

fragmentation during OA oxidation, where the highly aged and low volatile OA may dissociate

into more volatile fragments that are still highly functionalized and hygroscopic. In all, OA

hygroscopicity under sub-saturated conditions can be largely influenced by solubility,

molecular weight, molecular functional groups, and carbon number (Cai et al., 2021; Kuang et

al., 2020; Rastak et al., 2017; Rickards et al., 2013; Suda et al., 2012); to better understand the

variation of $\kappa_{\mathrm{OA}}$, more molecular investigations are needed.



In comparison with other campaigns, we find the variation of aerosol hygroscopicity in
the SEA is mainly due to changes in chemical composition, particularly sulphate and OA, as
well as variations in OA hygroscopicity during transport. To quantitatively investigate this
relationship, we came up with a parameterization using Fo, BCr, and $\kappa_{OA}$, and the $f(80\%)$ from
Mie simulations for internally mixed OA-$(NH_4)_2SO_4$-BC mixture with PNSD ($D_{gn}$=150 nm
and $\sigma_{sg}$=1.6). This suggests that the internal mixture of OA-$(NH_4)_2SO_4$-BC is a good
approximation of aerosols with respect to the $f(RH)$ prediction in 2016 and 2018 ORACLES
campaign.
Sensitivity study indicates that solely due to the increase in OA hygroscopicity
observed in our study, the aerosol scattering coefficients at 80 % RH can be amplified by 80 %.
Relying on the campaign's mean $\kappa_{OA}$ value leads to a poor prediction of $f(80\%)$. The
dependence of $f(RH)$ on $\kappa_{OA}$ suggests that using a constant $\kappa_{OA}$ can be acceptable when the OA
fraction is low and $\kappa_{OA}$ demonstrates limited variations. However, in situations where these
two conditions are not met, $\kappa_{OA}$ can significantly influence the scattering coefficients and thus
aerosol radiative effect. Therefore, accommodating the variability of $\kappa_{OA}$ is advisable.

*Competing interests.* At least one of the (co-)authors is a guest member of the editorial board
of Atmospheric Chemistry and Physics for the special issue "New observations and related
modelling studies of the aerosol-cloud-climate system in the Southeast Atlantic and southern
Africa regions". The authors have no other competing interests to declare.

*Special issue statement.* This article is part of the special issue "New observations and related
modeling studies of the aerosol-cloud-climate system in the Southeast Atlantic and southern
Africa regions (ACP/AMT inter-journal SI)". It is not associated with a conference.



*Data Availability.* Data sets are publicly available via the digital object identifier provided
under          ORACLES          Science          Team          reference:
https://doi.org/10.5067/Suborbital/ORACLES/P3/2018_V2.

*Acknowledgments.* The authors would like to thank the ORACLES team. Lu Zhang thanks the
postdoctoral fellowship funding from Tel Aviv University, Department of Exact Sciences.
Michal Segal-Rozenhaimer and Haochi Che are supported by United States Department of
Energy Atmospheric System Research (ASR) grant DE-SC0020084. Caroline Dang thanks the
NASA Postdoctoral Fellowship Grant. Paola Formenti is supported by the AErosols, RadiatiOn
and CLOuds in southern Africa (AEROCLO-sA) project funded by the French National
Research Agency under grant agreement n° ANR-15-CE01-0014-01, the French national
programs LEFE/INSU and PNTS, the French National Agency for Space Studies (CNES), the
European Union's 7th Framework Programme (FP7/2014-2018) under EUFAR2 contract
n°312609, and the South African National Research Foundation (NRF) under grant UID
105958. The authors thank Paul Zieger for useful comments on this article.

*Financial support.* This research has been supported by the Tel Aviv University (postdoctoral
fellowship); the United States Department of Energy (DOE) Atmospheric System Research
(ASR; grant DE-SC0020084); a NASA postdoctoral fellowship; the AErosols, RadiatiOn and
CLOuds in southern Africa (AEROCLO-sA) project funded by the French National Research
Agency (grant agreement no. ANR-15-CE01-0014-01); the French national programs
LEFE/INSU and PNTS; the French National Agency for Space Studies (CNES); the European
Union's Seventh Framework Programme (FP7/2014-2018; EUFAR2 contract no. 312609); and
the South African National Research Foundation (NRF; grant UID 105958).



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
