# Peer review of "Aerosol hygroscopicity over the South-East Atlantic Ocean during the"

_EGUsphere, 2023_

## Author Comment (AC1)

**Response to Reviewers** of egusphere-2023-2199 "Aerosol hygroscopicity over the South-East Atlantic Ocean during the biomass burning season: Part I – From the perspective of scattering enhancement" by L. Zhang et al.

We appreciate the time the reviewers have invested in assessing this manuscript. We have carefully considered each comment and have made revisions accordingly. Below, we address each point raised by the reviewers, with their comments presented in blue, our responses in red, and the corresponding manuscript text in black where applicable.

**Reviewer 1:**

This paper describes the vertical profile of aerosol hygroscopicity from the African continent during biomass burning season and the sensitivity of the observed hygroscopicity to aerosol chemical composition. The primary measurements used were made during two airborne field campaigns in 2016 and 2018. A main point of the paper is that when considering aerosol hygroscopicity it is important to realize that organic aerosol may have different water uptake values.

I think this paper is a useful addition to the hygroscopicity literature - however it would benefit from some more details about the measurements in order to better make their point. There is also some key information that should be mentioned sooner (e.g., PNSD was not important for f(RH) for this study, coarse mode was not important for this study.)

Comments/questions

1. line 80 - Suggests that organics can have 'kappa>1'? should this say '...kappa approaching 1.0'?

We have changed ">1" to "approaching 1.0".

2. line 113 - What was size cut for nephelometers?

Both M903 nephelometers were situated behind the solid diffuser inlet, which was designed to efficiently transmit particles with dry diameters up to 4.0 µm (McNaughton et al., 2007; Dobracki et al., 2023). We have included language on line 113 (original document) about the inlet and size cut: "Aerosol particles were introduced into the P3 via the solid diffuser inlet. The inlet was operated isokinetically by matching the flow rate to the external air flow velocity to within 5% (Dobracki et al., 2023). This inlet was designed to effectively transfer particles up to 4.0 µm dry diameter (McNaughton et al., 2007). The inner pipework was designed for minimal transport losses for particles up to 4.0 µm using an online particle loss calculator (Aerosol Calculator, https://tsi.com/getmedia/540a30fa-8444-49f6-814f-891495c70aa1/Aerocalc2001_1)."

3. Were nephelometer measurements corrected for truncation? Truncation corrections can be large when coarse mode aerosol are sampled.

Yes, we have applied the truncation correction for the nephelometer measurements, following the method in Anderson and Ogren (1998). We have included a sentence in the manuscript (line 120, original document) to clarify this: "Truncation correction has been performed for both Nephs according to Anderson and Ogren (1998)." Additionally, we have analysed the contribution of super-micron particles to the total scattering. Our analysis indicated that the contribution of these

super-micron particles is minimal and therefore can be neglected. We have added a new section in the supplement: S1. Minor contribution of super-micron particles to the total scattering. Please refer to the response to comment 10 for detailed wording.

4. line 116-117 - Where were RH sensors for RR neph located? Were they calibrated? (see, for example, description of RH sensor locations and calibration in manuscript cited reference Day et al., 2006).

The humidified Neph was situated downstream of a humidifier, which maintained the RH at the inlet of the Neph at around 80%. A small RH change within the Neph has been achieved; more details can be found in Howell et al. (2006). The RH probes in M903 have been corrected based on lab calibrations. We have rewritten the description of the humidified Neph: "Two Radiance Research M903 integrating nephelometers (Nephs) were operating in parallel, one (referred to as the 'reference Neph') under relatively dry conditions and the other (known as the 'humidified Neph') maintained at ~80 % RH. Particles entering the reference Neph were heated to the aircraft cabin temperature, significantly reducing their RHs in the Neph and resulting in most particles having an RH below 35 %. The humidified Neph was situated downstream of a humidifier, which maintained the RH at the inlet of the Neph at ~80 within a few percent, as detailed in Howell et al. (2006). The RH probes in M903 have been corrected based on lab calibrations and the RH errors are roughly 3 %. The temperature errors are about 0.5°C. Measurements were reported at 1 Hz. For the calculation of $f$(RH), data with a reference Neph RH greater than 35 % or a humidified Neph RH smaller than 76 % were excluded. The distribution of the RHs of both the reference and humidified Nephs used in this study are shown in Fig. S3 in the supplementary material. Calibrations were performed in the field with refrigerant R-134A (1,1,1,2-tetrafluoroethane). Truncation correction has been performed for both Nephs according to Anderson and Ogren (1998). All scattering coefficients and scattering enhancement factors are reported at 540 nm wavelength." The plot of Figure S3 is shown in the response to comment 5.

5. line 119 - Provide a probability distribution plot of ref RH and wet RH (it could go in supplemental).

We have added a plot of the RHs of reference Neph and humidified Neph in the supplement. In the text, we have added language: "The distribution of the RHs of both the reference and humidified Nephs used in this study are shown in Fig. S3 in the supplementary material."

[Figure]

"

Figure S3. Distributions of the RHs for reference and humidified Nephs during the campaign."

6. line 124 - What is largest diameter measured by AMS? did AMS provide a measure of sea salt?

The AMS measures the real-time non-refractory chemical speciation and mass loading as a function of particle size of fine aerosol particles with aerodynamic diameters between approximately 70 and 700 nm. Though there are small sea salt particles, they are largely super-micron and therefore cannot be detected by AMS. AMS can detect submicron Cl⁻. However, we have neglected it in our analysis due to its small concentration which accounts for less than 1±1% mass fraction (Fig. R1).

We have analysed the contribution of super-micron particles to the total scattering and found their contribution is less than 1.5% in 85% of the cases, which can be considered negligible. It means that if there are super-micron sea salt particles, their contribution would be equal to or smaller than this value and therefore can be neglected as well. We have added a more detailed discussion of the minor contribution of the super-micron particles in the main text and supplement, which can be found in detail in the response to comment 10.

[Figure]

Figure R1. The average vertical distribution of the mass ratio of chemical compositions from AMS and SP2 in every 200 m in the 2016 and 2018 ORACLES campaigns, respectively. The lines are the mean value in every 200 m bin. Errorbars and grey shading represent the standard deviation in every 200 m bin.

7. Line 129-132 - Did you perform a scattering closure study using the PNSD, refractive index from chemistry and the nephelometer measurements?

Yes, we have done a closure study of the dry scattering coefficient using the PNSD from UHSAS, the refractive index calculated from the chemical composition measured by AMS, and scattering coefficients from the reference Neph. The calculated scattering coefficients agree well with the measurements. We have added a scatter plot of the calculated and measured scattering coefficients under dry conditions in the supplement as below:

[Figure]

Figure S4. Calculated and measured scattering coefficients under dry conditions colored by the count of data in each dot.

We have added language describing this agreement in the manuscript: "Good agreement has been achieved for calculated and measured scattering coefficients under dry conditions, which indicates good data quality and provides the basis for the calculation of $f$(RH) and retrieval of $\kappa_{f(RH)}$. The comparison between calculated and measured scattering coefficients is shown in Fig. S4 in the supplement."

8. line 145 - Approximately 134 flight hours - is that before or after the constraints (dryRH<30%, scat>10 Mm-1 and f60 > 0.003 are applied? What was the final number of data points/hours studied?

The 134 flight hours are the final number of hours that are studied. We have made it clear by adding the language: "after applying the abovementioned constraints".

9. line 159 - This would be a good place to show that scattering closure was obtained for dry scattering and the PNSD. (it could go in supplemental materials). Relatedly - is assumed PNSD similar to measured PNSD - can you show that (e.g., a plot in supplemental materials).

We calculated the scattering coefficients using Mie model with inputs being the measured PNSD from UHSAS and the refractive index derived from chemical compositions provided by AMS. The calculated and measured scattering coefficients show very good agreement, which indicates good data quality and consistency among measurements from various instruments. It also provides the basis for the calculation of $f$(RH). We have added a scattering plot of the calculated and measured scattering coefficients in the supplement, which can be found in the response to comment 8.

The comment to PNSD is related to comments 25 and 29. We will have a detailed response there. The measured PNSD is used for the dry scattering coefficient closure study and the calculation of $\kappa_{f(RH)}$. The parameterization is based on the results obtained with the assumed PNSD. We used an

assumed PNSD instead of the measured ones for two reasons. One is that we want the parameterization to be widely applicable, not only for the specific PNSD. The other is that the scattering enhancement factor is not sensitive to PNSD. We have performed a sensitivity study evaluating the sensitivity of $f$(RH) to PNSD, which is shown in Section S2 in the Supplement. We have revised the method part and discussed the minor influence of PNSD to $f$(RH) there. Section 2.3 now reads:

"2.3 Modeling of $f$(RH)

The $f$(RH) can be modeled with the Mie theory (Mie, 1908). The Python package PyMieScatt (Sumlin et al., 2018), an implementation of the Mie theory, was applied in this study. Inputs of PyMieScatt include PNSD and complex refractive index. Dry particles beyond PM$_1$ (particulate matter with an aerodynamic diameter less than 1 μm) are not included in this calculation, supported by their minor contribution to the total scattering, as discussed in Section S1 of the supplement. A volume mixing rule was used to calculate the refractive index. The volume of inorganic salts was converted from those of $SO_4^{2-}$, $NO_3^-$, and $NH_4^+$ from AMS following a modified ion-pairing scheme (Gysel et al., 2007; Zhang et al., 2022). Good agreement has been achieved for calculated and measured scattering coefficients under dry conditions, which indicates good data quality and provides the basis for calculating $f$(RH) and retrieving $\kappa_{f(RH)}$. The comparison between calculated and measured scattering coefficients is shown in Fig. S4 in the supplement. By combining Mie model with the $\kappa$-Köhler theory, we can then calculate the scattering coefficients under humidified RH conditions. For more details of the calculation, refer to Zieger et al. (2013). Subsequently, $f$(RH) and γ can be obtained using Eq. 1 and 2. The theoretically calculated $f$(RH) in Sections 3.3.1 and 3.3.2 used an assumed PNSD and different chemical composition combinations. One assumed PNSD was used in these calculations due to its minor impact on $f$(RH), which has been discussed in detail in Section S2 in the supplement."

10. line 162-163 - 'particles beyond pm1 not included in calculation' - does this mean APS data is not used since UHSAS size range goes up to 1um? or was some sort of merging of the two size distributions done (for example, as described in Hand and Kreidenweis, 2002). Perhaps show plot(s) of full size distribution in supplement so can refer reader to it when say super-um particles made minimal contribution.

Yes, the UHSAS measures the particles with a diameter ranging from approximately 90 nm to 1 μm. There are two reasons that we neglected particles above 1 μm. Firstly, there are not many particles beyond 1 μm, supported by their minor contribution to the total scattering coefficients. Specifically, we have calculated the ratio of scattering contributed by supermicron particles to the total scattering and found that data points with this ratio below 0.015 account for 85% of the total data. A possibility distribution of this ratio is shown in Figure S1. Secondly, the $f$(RH) is not sensitive to PNSD as discussed in Section S2 in the Supplement.

We understand that it can be confusing that in the instrumentation part, we described in detail the merging of PNSDs from UHSAS and APS, while in the following analysis, we neglected the super-micron particles. Therefore we revised the language in the instrumentation part: "The dry particle number size distribution (PNSD) of PM$_1$ was provided by an ultra-high-sensitivity aerosol spectrometer (UHSAS). The UHSAS was calibrated with polystyrene latex (PSL) spheres, whose real refractive index n is 1.572 at the UHSAS laser wavelength (Howell et al., 2021). The UHSAS undersized particles in BB plumes; the undersized data were corrected according to Howell et al. (2021). The PNSD of super-micron particles was measured by an aerodynamic particle sizer (APS). The aerodynamic diameter of APS was converted to the volume equivalent diameter according to

DeCarlo et al. (2004). Particles were assumed to be spherical (shape factor = 1) with a density of 1.5 g cm$^{-3}$. However, since the super-micron particles made a minimal contribution to the total scattering coefficient, we have neglected the super-micron particles, and only UHSAS measurements are used in this study. The minor contribution of super-micron particles to the total scattering coefficients is described and illustrated in Section S1 and Fig. S1 in the supplement."

We have added a detailed discussion of the small contribution of supermicron particles in the supplement:

"S1. Minor contribution of super-micron particles to the total scattering

The PNSD of super-micron particles was measured by an aerodynamic particle sizer (APS), whose aerodynamic diameter was converted to the volume equivalent diameter according to DeCarlo et al. (2004). Particles were assumed to be spherical (shape factor = 1) with a density of 1.5 g cm$^{-3}$. The dry total scattering coefficients at 540 nm were measured by Radiance Research M903 integrating nephelometer.

We calculated the scattering coefficient of super-micron particles at 540 nm using the Mie model, with inputs being the PNSD from APS and a refractive index of 1.51+0.0048i for dust (Di Biagio et al., 2019). We have also calculated the scattering coefficient using the refractive index of sea salt. Since the results are similar to those using the refractive index of dust, only results from dust are shown here. The ratio of the scattering coefficient of super-micron particles to the total scattering coefficient reflects the contribution of super-micron particles to the total scattering. The distribution of this ratio is shown in Fig. S1. As illustrated in the figure, 85% of the data have a ratio of less than 0.015, indicating that the contribution of super-micron particles to the total scattering is less than 1.5% for 85% of the cases, demonstrating the minimal impact of super-micron particles on the total scattering.

[Figure]

Figure S1. The PDF distribution of the ratio of the scattering coefficient of super-micron particles to the total scattering coefficient."

11. line 168 - Is there no information on contribution of sea salt? It seems like sea salt should be considered since over ocean and/or discuss why it's ok not to consider it. Later on (line 201) mentions the marine boundary layer so it seems there would be some influence of sea salt on the measurements

This comment is related to comments 6 and 10. The sea salt particles are largely super-micron, though there can be sub-micron sea salt particles. To assess the contribution of super-micron particles to the total scattering, we calculated the scattering coefficient contributed by those particles using the Mie model. Results show that for 85% of the cases, this contribution is less than 1.5%, indicating a minor contribution from super-micron particles to the total scattering. This analysis has been added to the manuscript (also see response to comment 10). We have also investigated submicron $Cl^-$-containing particles and found their contribution is negligible. As a result, we have not included $Cl^-$ in our study. Please also refer to the response to comment 6 for more detail. We have added language explaining the neglect of $Cl^-$ in submicron particles in line in line 278 the original text: "We neglected chloride in this study as it accounts for less than 1±1% mass fraction."

12. Figure 2 - There appears to be an inverse relationship between OA/BC and plume age. Could/should note this when discussing the figure.

We have mentioned this inverse relationship in the original line 228: In 2018, temperature and plume age are closely correlated (Pearson correlation coefficient of 0.51), and the decrease in OA/BC is accompanied by ageing (Pearson correlation coefficient of 0.57), as shown in Fig. 2a and b. Following the reviewer's comment, we have emphasized this relationship by adding a description of this inverse relationship between OA/BC and plume age in line 213 in the original manuscript: "In the meanwhile, the OA/BC shows a clear reverse trend with the plume age in 2018, this inverse relationship is less obvious in 2016."

13. line 195 - Sentence refers to 'figure 1a' --> change to figure 1 because there is only 1 pane in figure 1.

Done.

14. line 195 - Is there something in figure 1 that indicates subsidence? Please elaborate! Also please label countries so when refer to Namibian coast reader can see where you are talking about.

The positive values of omega, i.e. 55 and 65 $hPad^{-1}$ represent the subsidence. It is illustrated in the figure caption, "Solid and dashed lines represent the subsidence of 55 and 65 hectopascals per day (hPa d-1)." It can be seen from Fig. 1 that there is a strong subsidence off the Namibian coast. However, to make it clear, we have rewritten the sentence as: "aerosols are less aged than those in the 2018 campaign due to the subsidence (positive values of omega) near the Namibian coast (Fig. 1)."

The revised figure is shown below: "

[Figure]

Figure 1. Flight tracks in 2016 and 2018 ORACLES campaigns. Map of October mean of ERA5 600 hPa RH overlaid by the 600 hPa zonal wind (purple contours; 6, 7, and 8 m s$^{-1}$), 600 hPa horizontal wind vector (purple arrows; m s$^{-1}$), and ORACLES flight tracks in 2016 (yellow) and 2018 (blue), respectively. White contours are the 2016 September mean vertical velocity, omega, at 800 hPa. Solid and dashed lines represent the subsidence of 55 and 65 hectopascals per day (hPa d$^{-1}$). "

Following the reviewer's comment, we have added the country border in Fig.1 to make it clear.

15. line 196 &197 - Do you have a citation for the cloud top height information? Please provide.
We have added a citation from Redemann et al. (2021).

16. line 211 - The authors note that the OA/BC ratio 'removes the dilution effect during transport'. but they should also note that it's useful for looking at because it can indicate something about processing as discussed later - e.g., gas phase organics condensing on existing particles or loss of hygroscopic particles due to wet scavenging.
We have added language about the processing in line 211 (original text): "The OA/BC ratio, representing the OA mass concentration normalised by that of BC to remove the dilution effect during transport and an indication of OA processing".

17. line 218 - Mentions Figure 3 only considers measurements above 1.4 km to minimize marine influence. Why is 1.4 km the chosen altitude for that? Should all analyses/plots use that altitude constraint? Figure 2 goes down to 1 and 0.5 km for the two different years. Should a line be added to figure 2 at 1.4 km with a note indicating above that height is considered above the MBL?
We applied this altitude constraint to analyze OA processing to minimise the marine influence. We have not applied this constraint to the aerosol hygroscopicity analysis because aerosol hygroscopicity largely depends on chemical compositions despite their various processing before. The 1.4 km is the maximum height of the MBL for both years, above which is not considered

MBL. We have added a line at 1.4 km in Fig. 2 following the reviewer's advice, and also noted that above this height is not considered the MBL in the figure caption: "

[Figure]

Figure 2. The vertical distribution of plume age and chemical composition. (a, c) Variation of plume age (black), OA/BC, and SO4/BC with altitude in 2016 (upper) and 2018 (lower) ORACLES campaigns, respectively. Grey dots show the distribution of plume age with the altitude. (b, d) The average vertical distribution of the mass ratio of chemical compositions and the average mass concentration of $PM_1$ from AMS and SP2 in every 200 m in 2016 and 2018 ORACLES campaigns, respectively. The lines are the mean value in every 200 m bin. Errorbars and grey shading represent the standard deviation in every 200 m bin. The red dashed lines show the maximum height of the MBL during the study period. "

18. Figure 5 - Perhaps say shading indicates 10 and 90 percentiles rather than dashed lines since the line for 2016 mean is also dashed. Also, you should NOT combine mean with percentiles - that is mixing statistical parameters. Either use median and percentiles or

mean and standard deviation! Are the f(RH) values (and gamma values) above and below 2km statistically different (e.g. can you use something like a student t-test or some other appropriate statistical test to say how different they are at what level of confidence)?

We have changed the mean to the median, to make it statistically consistent with the 10 and 90 percentiles. The revised plot is shown below.

[Figure]

We have revised the figure caption to: "Figure 5. Vertical profiles and PDF of $f(80\%)$ (a, d), $\kappa_{f(RH)}$ (b, e), and $\kappa_{OA}$ (c, f) for aerosols in the 2016 (dotted line) and 2018 (solid line) ORACLES campaign. The lines in a, b, and c represent the medians, and the shadings in a, b, and c represent the $10^{th}$ and $90^{th}$ percentiles."

We have performed Levene's test for medians for $f(80\%)$, $\kappa_{f(RH)}$, and $\kappa_{OA}$, and all the p-values are below 0.05, indicating that $f(80\%)$, $\kappa_{f(RH)}$, and $\kappa_{OA}$ are statistically different above and below 2 km, with a confidence level of 95%. We have revised the sentence in line 285 as: "The results from the Levene's test for medians for $f(80\%)$, $\kappa_{f(RH)}$, and $\kappa_{OA}$ indicate that $f(80\%)$, $\kappa_{f(RH)}$, and $\kappa_{OA}$ are statistically different above and below 2 km, with a confidence level of 95%."

19. line 290 Change 'diameter' --> 'value' (in both places on the line)
Done. The sentence now reads: "For $f(80\%)$ below 2 km, a primary mode with a value of around 1.45 is evident, but there is also a second mode with a value of around 1.81 for aerosols in both years."

20. lines 296-313 - Put comparisons in a table and/or do similar figure as figure 3 in Titos et al. (2016) rather than this paragraph which is rather long and hard to follow.
We have added a table comparing the hygroscopicity of biomass burning aerosols. This paragraph has been rewritten to make it concise and more readable. It now reads: "For aerosols above 2 km, the mean and standard deviation of $f(80\%)$ and $\kappa_{f(RH)}$ are 1.40±0.17 and 0.19±0.07, respectively (Fig. 5, Table 1, and Table S2). These values indicate less hygroscopic particles (Liu et al., 2011) and are lower than those for marine aerosols (Zieger et al., 2010; Carrico et al., 2003) but higher

than those for dust and polluted dust particles (Bukowiecki et al., 2016; Zhang et al., 2015a). They are comparable to smoke-dominated aerosols, such as the smoke from savanna fires in Australia (Gras et al., 1999) and the BBAs from forest fires in the northeast US (Wang et al., 2007). These values are slightly higher than the $f$(80%) in Brazil (SCAR-B) (Kotchenruther and Hobbs, 1998). The particles below 2 km are more hygroscopic (Liu et al., 2011). The mean and standard deviation of $f$(80%) and $\kappa_{f(RH)}$ are 1.51±0.22 and 0.23±0.08, respectively, placing them in the upper ranges of BBA hygroscopicity reported in the literature. These values are comparable to those of the aged smoke in Africa (SAFARI, Magi and Hobbs, 2003) and the Yangtze River Delta background station (Zhang et al., 2015a). They match the $\kappa_{f(RH)}$ of 0.22 at a rural site in southern China (Kuang et al., 2021) but are lower than the values for BBAs in East Asia (ACE-Asia, Kim et al., 2006) and agricultural burning in INDOEX (Indian Ocean Experiment, Sheridan et al., 2002).

Table 1. The $f$(RH) of biomass burning aerosol from the literature.

| $f$(RH) | RH | Location | Fuel type and notes | Reference |
|---|---|---|---|---|
| 1.37 | 80% | Australia | light-wooded savanna fires | Gras et al., 1999 |
| 1.40 | 82% | northeast US | forest fires | Wang et al., 2007 |
| 1.16 | 80% | Brazil (SCAR-B[a]) | grass, shrub, and trees | Kotchenruther and Hobbs, 1998 |
| 1.44 ± 0.02 | 80% | Southern Africa (SAFARI 2000[b]) | aged heavy smoke | Magi and Hobbs, 2003 |
| 1.60±0.20 | 85% | Korea (ACE-Asia[c]) | BBAs | Kim et al., 2006 |
| 1.58±0.21 | 85% | India Ocean (INDOEX[d]) | agricultural burning | Sheridan et al., 2002 |
| 1.51±0.22 | 80% | South-East Atlantic Ocean (below 2 km, ORACLES) | savanna[e] | This study |
| 1.40±0.17 | 80% | South-East Atlantic Ocean (above 2 km, ORACLES) | savanna[e] | This study |

[a] Smoke, Clouds, and Radiation-Brazil
[b] Southern African Regional Science Initiative 2000
[c] Aerosol Characterization Experiment
[d] Indian Ocean Experiment
[e] Che et al., 2022"

21. line 324-325 - Pearson correlation coefficient of -0.35. so R2 ~ 0.13. Is this statistically significant?

It is not statistically significant. The Pearson correlation coefficient of -0.35 suggests a weak to moderate inverse linear relationship between $\kappa_{OA}$ and OSc. Therefore, we described this as a slight increase in the manuscript.

22. Fig 6 - I don't see dotted gray lines (except for gridlines). There is also too much information on this plot. I suggest making two plots one with measurement points and one with model results maybe with ACE-Asia fit on both plots for ease of comparability.

We want to demonstrate the good overlap of measurements from various campaigns and this study and the good agreement between these measurements and theoretical calculations. Therefore, it is difficult to separate this plot. However, we recognize the importance of clearly displaying the 95% prediction bands of ACE-Asia, as these generally provide an outer boundary for measurements from different campaigns. Thus, we have replotted this figure, adjusting the colors and line types. We hope this revision makes the figure clearer. The revised plot and caption are shown below: "

[Figure]

Figure 6. $\gamma$ versus Fo in various campaigns and for internally mixed OA-$(NH_4)_2SO_4$-BC mixtures. Fo represents the ratio of mass concentrations of OA to OA and $SO_4^{2-}$. Solid lines in light blue and red represent the linear fits for ACE-Asia and ORACLES, respectively. Solid blue lines show the 95% prediction bands for the ACE-Asia data, in light blue rectangles, taken from Quinn et al. (2005). Colorbar represents the plume age (days) in ORACLES. Data for SEAC[4]RS is shown by dark blue diamond, taken from Shingler et al. (2016). DISCOVER-AQ data is shown by yellow diamonds, taken from NASA Langley Research Center Atmospheric Science Data Center (Atmospheric Science Data Center, 2015). Fitting lines for two European sites Melpitz (solid orange line) and Hyytiälä (solid black line) are from Zieger et al. (2015). Blue, green, and purple lines represent results for internally mixed OA-$(NH_4)_2SO_4$-BC mixtures with 1) a range of BC mass fraction (BCr, solid for 5% and dashed for 25%) and 2) OA with $\kappa_{OA}$ of 0 (blue), 0.2 (green), and 0.6 (purple) from Mie calculations assuming a lognormal size distribution with a geometric mean diameter $D_{gn}$ of 150 nm and a standard deviation $\sigma_{sg}$ of 1.6. "

23. line 376 "was calculated following the lognormally distribution" I'm not sure what you are saying here. I think you mean assuming the PNSD is lognormally distributed? Do the assumed values of Dg and sigma match the measured PNSD?

We are sorry that we have not expressed this clearly, we mean the PNSD was calculated assuming it is lognormally distributed. The geometric mean diameter ($D_{gn}$) was assumed to be 150 nm and the standard deviation ($\sigma_{sg}$) to be 1.6. Then we applied this PNSD in the calculation of the scattering enhancement factor. We have analyzed in the supplementary that PNSD only has a minor influence on the calculation of $f$(RH). Please refer to the response to comment 9 in more detail. We have rephrased this sentence as: "was calculated assuming it is lognormally distributed, with the geometric mean diameter ($D_{gn}$) of 150 nm and the standard deviation ($\sigma_{sg}$) of 1.6."

24. line 377 - Caption for figure 6 says sigma=1.5, this line says sigma=1.6

There is a typo in the caption which should be 1.6. We have corrected it and the caption now reads: "Blue, green, and purple lines represent results for internally mixed OA-$(NH_4)_2SO_4$-BC mixtures with 1) a range of BC mass fraction (BCr, solid for 5% and dashed for 25%) and 2) OA with $\kappa_{OA}$ of 0 (blue), 0.2 (green), and 0.6 (purple) from Mie calculations assuming a lognormal size distribution with a geometric mean diameter $D_{gn}$ of 150 nm and a standard deviation $\sigma_{sg}$ of 1.6. "
We have checked through the manuscript to ensure the sigma is always 1.6.

25. lines 381-382 How much variation was there in the measured PNSD temporally and spatially (horizontal and vertical)? How does assuming a constant size distribution affect your results? Very possibly I am missing something, but it seems like if a constant size distribution is assumed, then by definition the hygroscopicity will depend on composition. It might be good to discuss in methods section more about the size distribution - characteristics such as Dg and sigma as well as spatial/temporal changes.

We have analyzed the sensitivity of $f$(RH) to PNSD in the supplementary and found that PNSD only has a minor influence on the calculation of f(RH). The chemical composition and the $\kappa_{OA}$ largely determine the aerosol hygroscopicity. Therefore, we have not included much discussion on the variation of PNSD in this study. However, as the reviewer suggested in the comment 29, we should have mentioned the minor impact of PNSD on $f$(RH) sooner in the main text. We have revised the method section and discussed the impact of PNSD on $f$(RH) there, for more details, please refer to the response to comment 9.

26. line 398 - Has sigma=1.6 also instead of 1.5. Use consistent sigma for all calculations or explain why not doing so?

The sigma is 1.6; we have corrected it.

27. line 399 - How do results change if RHref = 30% is used? What is the median value of the RHref of your measurements?

The median value of the RHref is 12 %. We have added a figure of the distributions of the RHs of both Neph following the reviewer's advice.

[Figure]

Figure S3. Distributions of the RHs for reference and humidified Nephs during the campaign.

In the theoretical calculation of $f$(RH), the reference RH of 0 is commonly used to ensure that the particles are completely dry. If we change it to 30%, the $f$(RH) can be a bit lower. We chose to use 0 as the reference RH to keep consistency with previous studies and also because it is closer to our measurements.

28. line 401 Change 'all with a span of 0.02' --> 'all in increments of 0.02'
Done.

29. line 424-425 - This is the first time it is mentioned that the effect of PNSD on f(RH) is small. That should be mentioned much sooner and perhaps with a little more detail in the main text.

We agree that moving this part earlier would make it easier to read. We have revised the method part and moved it to Section 2.3. The revised Section 2.3 can be found in the response to comment 9.

30. line 405-415 - Do the 2nd and 5th order polynomials have a physical interpretation? Do you expect such equations to be widely applicable or only for the special case (BBA over SEA) studied here?
It would be best to have a physical interpretation of the 2nd- and 5th-order polynomials, however, we have not found a physical interpretation of this parameterization. It is only a statistical empirical parameterization. This parameterization is based on theory, summarizing the results of $f$(RH) calculated from the assumed PNSD and various chemical composition combinations. The PNSD is assumed to be lognormally distributed with a geometric mean diameter ($D_{gn}$) of 150 nm and a standard deviation ($\sigma_{sg}$) of 1.6. However, PNSD only has a minor influence on $f$(RH), therefore won't affect its application. The chemical composition and $\kappa_{OA}$ are inputs, also not limited to special cases. Therefore, we expect a widespread application of this parameterization.

31. line 448 - How is the deviation of f(80%) calculated? Why is the deviation always positive?
We have added the calculation of the deviation of $f$(80%) both in the main text and the caption of Fig. 9: "The deviation of $f$(80%) was calculated as $f$(80%, $\kappa_{OA}$)-$f$(80%, $\kappa_{OA}$=0)."

32. line 455 - Change 'As well' --> 'Additionally'
Done.

33. line 463-468 - The discussion of the North China Plain data doesn't add anything. You've already made the point that KappaOA and Fo need to represent the actual data to get a good f(RH) value. I would remove this data and discussion.
We have removed the North China Plain from both the plot and text. The figure now looks: "

[Figure]

Figure 9. Sensitivity of $f(80\%)$ to $\kappa_{OA}$. The deviation of $f(80\%)$ was calculated as $f(80\%, \kappa_{OA}=0)$-$f(80\%, \kappa_{OA})$. The OA to OA + SO4 ratio (Fo) is represented by the colorbar. The blue and orange lines represent the variation at $75^{th}$ and $25^{th}$ percentile of Fo in both years' ORACLES campaign, respectively."

34. line 495 - Give median to go with percentiles.
We have replaced the standard deviation with the percentile.

35. Line 512 - Change 'Sensitivity study' --> 'A sensitivity study'
Done.

Hand, J.L., Kreidenweis, S.M., 2002. A new method for retrieving particle refractive index and effective density from aerosol size distribution data. Aerosol Science & Technology, 36:10, 1012-1026.

**Reviewer 2:**

This paper presents an assessment of the controls of aerosol hygroscopicity for well-aged biomass burning aerosol over the South-East Atlantic ocean during ORACLES 2016 and 2018. Measurements of f(RH) and chemical composition ratios are shown in order to derive the hygroscopicity of organic aerosol specifically. A complicated parameterization is developed that depends on organic/sulfate ratio, black carbon mass fraction, and organic aerosol hygroscopicity, and is shown to reproduce the measured data. The conclusions are important: variability of the organic hygroscopicity of biomass burning can significantly alter the overall hygroscopicity of particles.

The manuscript is generally well-written and organized. More measurement detail is needed in the paragraph at line 111.

1. Please include a description about how the relative humidity (RH) sensor for the humidified nephelometer was calibrated, and if this is the basis for the stated 3% uncertainty. Also, please include a description of how the sample was humidified, or include a reference for the approach.

We have added a sentence on the humidification of aerosols, a description of the calibration of the RH sensor, and a reference from Howell et al. (2006) in the main text (original line 114). Now this part reads: "Two Radiance Research M903 integrating nephelometers (Nephs) were operating in parallel, one (referred to as the 'reference Neph') under relatively dry conditions and the other (known as the 'humidified Neph') maintained at ~80 % RH. Particles entering the reference Neph were heated to the aircraft cabin temperature, significantly reducing their RHs in the Neph and resulting in most particles having an RH below 35%. The humidified Neph was situated downstream of a humidifier, which maintained the RH at the inlet of the Neph at ~80% within a few percent, as detailed in Howell et al. (2006). The RH probes in M903 were corrected based on lab calibrations and the RH errors are roughly 3%. The temperature errors are about 0.5°C. Measurements were reported at 1 Hz. For the calculation of $f$(RH), data with a reference Neph RH greater than 35% or a humidified Neph RH smaller than 76% were excluded. The distribution of the RHs of both the reference and humidified Nephs used in this study are shown in Fig. S3 in the supplementary material."

2. For what size range are the nephelometers measuring scattering coefficient? Is this governed by the aircraft inlet or by internal tubing limitations?

Aerosol particles were brought into the P3 through the solid diffuser inlet (SDI), which has been shown to efficiently transmit particles at dry diameters up to 4.0 μm (McNaughton et al., 2007). Internal pipework was designed to minimize transport losses for particles up to 4.0 μm (Dobracki et al., 2023). We have added this information in line 113 of the original version of the manuscript: "Aerosol particles were introduced into the P3 via the solid diffuser inlet. The inlet was operated isokinetically by matching the flow rate to the external air flow velocity to within 5% (Dobracki et al., 2022). This inlet was designed to effectively transfer particles up to 4.0 μm dry diameter (McNaughton et al., 2007). The inner pipework was designed for minimal transport losses for particles up to 4.0 μm using an online particle loss calculator (Aerosol Calculator, https://tsi.com/getmedia/540a30fa-8444-49f6-814f-891495c70aa1/Aerocalc2001_1)."

3. **Are there any losses in the humidification system that would complicate a quantitative comparison of dry vs. humid scattering?**

The pipework has been designed to minimize particle loss, and we have added a reference to this in the text.

4. Additionally, only mass fractions of aerosol components are presented. Without presenting absolute concentrations, it is difficult to assess what components are driving changes in mass fraction, and if low concentrations are driving increased uncertainty in the derived ratios. Please consider including profiles with actual mass concentrations for the AMS components and BC.

We have added a vertical profile of the average mass concentration of $PM_1$ in the plot. The calculation of the mass concentration of $PM_1$ is added in the text (line 178, original manuscript): "The total volume of $PM_1$ is calculated as the sum of the inorganic salts and organics from AMS and the BC from SP2." The revised plot is shown below: "

[Figure]

Figure 2. The vertical distribution of plume age and chemical composition. (a, c) Variation of plume age (black), OA/BC, and SO4/BC with altitude in 2016 (upper) and 2018 (lower)

ORACLES campaigns, respectively. Grey dots show the distribution of plume age with the altitude. (b, d) The average vertical distribution of the mass ratio of chemical compositions and the average mass concentration of $PM_1$ from AMS and SP2 in every 200 m in 2016 and 2018 ORACLES campaigns, respectively. The lines are the mean value in every 200 m bin. Errorbars and grey shading represent the standard deviation in every 200 m bin. The red dashed lines at 1400 m show the maximum height of the MBL during the study period."

**5.131    How was the UHSAS calibrated?  PSL or at another refractive index?**

The UHSAS was calibrated with polystyrene latex (PSL) spheres (real refractive index n = 1.572 at the UHSAS laser wavelength) from 70 to 800 nm diameter. The details of the calibration are described by Howell et al. (2021). We have added and referenced this calibration method in the main text. The description of UHSAS now reads: "The dry particle number size distribution (PNSD) of $PM_1$ was provided by an ultra-high-sensitivity aerosol spectrometer (UHSAS). The UHSAS was calibrated with polystyrene latex (PSL) spheres, whose real refractive index n is 1.572 at the UHSAS laser wavelength (Howell et al., 2021). The UHSAS undersized particles in BB plumes; the undersized data were corrected according to Howell et al. (2021)."

**6.134    What particles are assumed to dominate the coarse mode to justify a spherical assumption?  Sea-salt or dust?**

We have neglected the supermicron particles due to their small concentration and negligible contribution to particle scattering and the scattering enhancement factor. In addition, the scattering enhancement factor is insensitive to PNSD, we have discussed it in Section 2 in the Supplement. We have added Section 1 in the Supplement to analyze this small contribution of supermicron particles to total particle scattering: "S1. Minor contribution of super-micron particles to the total scattering

The PNSD of super-micron particles was measured by an aerodynamic particle sizer (APS), whose aerodynamic diameter was converted to the volume equivalent diameter according to DeCarlo et al. (2004). Particles were assumed to be spherical (shape factor = 1) with a density of 1.5 g cm-3. The total scattering coefficients at 540 nm were measured by Radiance Research M903 integrating nephelometer.

We calculated the scattering coefficient of super-micron particles at 540 nm using the Mie model, with inputs being the PNSD from APS and a refractive index of 1.51+0.0048i for dust (Di Biagio et al., 2019). We have also calculated the scattering coefficient using the refractive index of sea salt and the result are similar to that using the refractive index of dust; therefore, not shown here. The ratio of the scattering coefficient of super-micron particles to the total scattering coefficient reflects the contribution of super-micron particles to the total scattering. The distribution of the ratio is shown in Fig. S1. There are 85% data with the ratio less than 0.015, i.e. the contribution of super-micron particles to the total scattering is less than 1.5%.

[Figure]

Figure S1. The PDF distribution of the ratio of the scattering coefficient of super-micron particles to the total scattering coefficient."

We have revised the language in the instrument section accordingly: "The dry particle number size distribution (PNSD) of $PM_1$ was provided by an ultra-high-sensitivity aerosol spectrometer (UHSAS). The UHSAS was calibrated with polystyrene latex (PSL) spheres, whose real refractive index n is 1.572 at the UHSAS laser wavelength (Howell et al., 2021). The UHSAS undersized particles in BB plumes; the undersized data were corrected according to Howell et al. (2021). The PNSD of super-micron particles was measured by an aerodynamic particle sizer (APS). The aerodynamic diameter of APS was converted to the volume equivalent diameter according to DeCarlo et al. (2004). Particles were assumed to be spherical (shape factor = 1) with a density of 1.5 g $cm^{-3}$. However, since the super-micron particles made a minimal contribution to the scattering coefficient, we have neglected the super-micron particles in this study and only UHSAS measurements are used. The minor contribution of super-micron particles to the total scattering coefficients is described and illustrated in Section S1 and Fig. S1 in the supplement."

7.135    What do the subscripts "i" and "X" represent in the equation? Are you assuming kappa values for BC, and what kappa values are assumed for inorganic components? Please include a better description of Equation (3), which is vital to the paper results.

There should be no "X" in the equation. We have rewritten the equation 3. The *inorg* represents inorganic salts, which were converted from $SO_4^{2-}$, $NO_3^-$, and $NH_4^+$ measured from AMS following a modified ion-pairing scheme (Gysel et al., 2007; Zhang et al., 2022). The *i* represents each inorganic salt. The refractive indices and kappa values of various inorganic salts, organics, and BC have been listed in Table S1 in the Supplement. We have described the equation in more detail and now it reads:

"Therefore, the hygroscopicity parameter of OA, $\kappa_{OA}$, can be calculated as:

$$\kappa_{OA} = \frac{\kappa_{f(RH)} - (\sum_{i=inorg} \kappa_i \varepsilon_i + \kappa_{BC} \varepsilon_{BC})}{\varepsilon_{OA}}, \tag{3}$$

where *inorg* represents inorganic salts, which were derived from the $SO_4^{2-}$, $NO_3^-$, and $NH_4^+$ ions measured from AMS following a modified ion-pairing scheme (Gysel et al., 2007; Zhang et al., 2022). The subscript $i$ denotes each individual inorganic salt. $\varepsilon$ represents the volume fraction of each component, calculated as the ratio of the volume of each component to the volume of PM$_1$. The PM$_1$ volume is computed as the sum of the volumes of inorganic salts, OA, and BC. The hygroscopic parameter $\kappa$ and density used in this study can be found in Table S1. "

    8.182      Since you are assessing hygroscopicity, and invoking the SO4/OA ratio to explain variability, can you plot the vertical distribution of SO4/OA?

This comment is linked to comment 14. We have added a vertical distribution of SO4/OA following the reviewer's comment. The wording has been added in line 286 in the original manuscript: "i.e. more sulfate and less OA at lower altitudes (Fig. S5)."

[Figure]

"
Figure S5. Vertical distributions of SO4/OA and f(80%) in 2016 and 2018 ORACLES campaigns. The lines are the mean value in every 200 m bin. Shadings represent the standard deviation in every 200 m bin. "

    9.191      You reference a pressure but are plotting altitude. Please be consistent.
We have reworded "600 – 700 hPa" to "3 – 4 km" to make it consistent.

    10. 194      The airmass age at the surface from Fig2 seems very similar between years. Both 10-12 days. Please confirm this statement. The biggest difference seems to be that the 2018 data looks less aged aloft.
We have drawn the plume ages of both campaigns as shown in Fig. R2. Flights in 2016 ORACLES (Fig. 1 in the manuscript, yellow lines) are in the region of 8-24° S and 0-15° E, traversing both the southern African Easterly Jet (AEJ-S) region and the continent anticyclone. As a result, aerosols at higher altitudes in 2016 ORACLES include both less-aged (<4 d) particles coming directly from the continent and highly aged (>10 d) particles transported from the west/north, resulting in a much larger variation of plume age in each level that that in 2018, which can be seen

from the larger shading area in 2016 in the figure below. At lower altitudes, aerosols are less aged than those in the 2018 campaign due to the subsidence (positive values of omega) near the Namibian coast (revised Fig. 1).

[Figure]

Figure. R2 Vertical distributions of the plume age in 2016 and 2018. The lines are the mean value in every 200 m bin. Shadings represent the standard deviation in every 200 m bin.
"

[Figure]

Figure 1. Flight tracks in 2016 and 2018 ORACLES campaigns. Map of October mean of ERA5 600 hPa RH overlaid by the 600 hPa zonal wind (purple contours; 6, 7, and 8 m s⁻¹), 600 hPa horizontal wind vector (purple arrows; m s⁻¹), and ORACLES flight tracks in 2016 (yellow) and 2018 (blue), respectively. White contours are the 2016 September mean vertical velocity, omega, at 800 hPa. Solid and dashed lines represent the subsidence of 55 and 65 hectopascals per day (hPa d⁻¹). "

11. 248      You are plotting and interpreting the dependence on ambient temperature; how does that temperature relate to the temperature directly at the AMS? During the description of the dry Nephelometer, you note that drying occurs because of heating inside the aircraft cabin, so does this also apply to the AMS measurements? Does the measurement temperature vary with the ambient temperature significantly?

The cabin temperature of NASA P-3 is held between 291-297K (Airborne Laboratory Experimenter Handbook, https://airbornescience.nasa.gov/sites/default/files/P-3B%20Experimenter%20Handbook%20548-HDBK-0001.pdf). The temperature can vary depending on the operating location and the location of heat-producing equipment. There is no direct temperature measurement at the inlet of AMS; however, we can have a general idea of the cabin temperature from the one that is measured at the outlet of M903 Neph (see figure below). The variation is within 10K, while the variation of the ambient temperature can reach up to 30K.

[Figure]

Fig. The temperature of the outlet of the dry neph (M903) vs. the ambient temperature.
We acknowledge that there can be losses during the transmission of particles through the inlet and inner pipe in the aircraft cabin. When ambient temperatures are low, particles entering the warmer cabin environment may experience the evaporation of volatile and semi-volatile compounds, leading to a reduction in particulate phase organic content and potentially resulting in an underestimation of organics by the AMS. This is a recognized issue in aerosol sampling, particularly in airborne studies, and remains challenging to quantify accurately.
Despite the difficulty in quantifying this loss, our study's f44 analysis and O:C ratios indicate that the organic aerosols over the Southeast Atlantic Ocean are low in volatility and highly oxidized, which suggests that any loss of organics back to the gas phase is minimal and certainly within the AMS's uncertainty limits. In addition, TDMA results show little variation in particle distribution when heated to 150°C, further supporting the conclusion that the temperature difference between the ambient environment and the cabin has a minor impact on particle partitioning.

12. 265      Is the AMS sensitive to sea-salt sulfate using a standard vaporizer temperature of 600C? Was the AMS operated at 600C?

The AMS operates at a high vaporizer temperature of 650°C, which is chosen as it is more effective for sampling organic aerosols during the ORACLES campaign. The AMS is designed to measure non-refractory submicron particulate matter. Submicron sea-salt sulfate particles can be measured by AMS, while sea-salt sulfate particles are usually large and most of them are supermicron, beyond the detection range of AMS. However, we have evaluated the influence of super-micron to aerosol scattering during our campaign and it is found that the impact of supermicron particles can be neglected in the study. We have added a section in the supplement analyzing this influence. Please refer to the response to comment 7 for the added section in the supplementary material.

13. 285      I do not see any plots to assess the vertical variability of SO4/OA to assess this statement.

We have added a plot showing the vertical distribution of SO4/OA in the supplement. Please refer to the response to comment 11 for more details.

14. 293      What is the typical MBL height during measurements? Is there a measurement to corroborate this statement?

The maximum MBL height is 1.4 km during the campaign. We have added a dashed red line in Figure 2 of the original manuscript to denote this height. Please refer to the response to comment 5 for the revised plot.

New references invoked within the response:

Anderson, T. L. and Ogren, J. A.: Determining Aerosol Radiative Properties Using the TSI 3563 Integrating Nephelometer, Aerosol Sci. Technol, 29, 57–69, https://doi.org/10.1080/02786829808965551, 1998.

Che, H., Segal-Rozenhaimer, M., Zhang, L., Dang, C., Zuidema, P., Sedlacek III, A. J., Zhang, X., and Flynn, C.: Seasonal variations in fire conditions are important drivers in the trend of aerosol optical properties over the south-eastern Atlantic, Atmospheric Chemistry and Physics, 22, 8767–8785, https://doi.org/10.5194/acp-22-8767-2022, 2022.

DeCarlo, P. F., Slowik, J. G., Worsnop, D. R., Davidovits, P., and Jimenez, J. L.: Particle Morphology and Density Characterization by Combined Mobility and Aerodynamic Diameter Measurements. Part 1: Theory, Aerosol Science and Technology, 38, 1185–1205, https://doi.org/10.1080/027868290903907, 2004.

Dobracki, A., Zuidema, P., Howell, S. G., Saide, P., Freitag, S., Aiken, A. C., Burton, S. P., Sedlacek III, A. J., Redemann, J., and Wood, R.: An attribution of the low single-scattering albedo of biomass burning aerosol over the southeastern Atlantic, Atmos. Chem. Phys., 23, 4775–4799, https://doi.org/10.5194/acp-23-4775-2023, 2023.

Howell, S. G., Clarke, A. D., Shinozuka, Y., Kapustin, V., McNaughton, C. S., Huebert, B. J., Doherty, S. J., and Anderson, T. L.: Influence of relative humidity upon pollution and dust during ACE-Asia: Size distributions and implications for optical properties, J. Geophys. Res., 111, 2004JD005759, https://doi.org/10.1029/2004JD005759, 2006.

McNaughton, C. S., Clarke, A. D., Howell, S. G., Pinkerton, M., Anderson, B., Thornhill, L., Hudgins, C., Winstead, E., Dibb, J. E., Scheuer, E., and Maring, H.: Results from the DC-8 Inlet Characterization Experiment (DICE): Airborne Versus Surface Sampling of Mineral Dust and Sea Salt Aerosols, Aerosol Science and Technology, 41, 136–159, https://doi.org/10.1080/02786820601118406, 2007.

Redemann, J., Wood, R., Zuidema, P., Doherty, S. J., Luna, B., LeBlanc, S. E., Diamond, M. S., Shinozuka, Y., Chang, I. Y., Ueyama, R., Pfister, L., Ryoo, J.-M., Dobracki, A. N., da Silva, A. M., Longo, K. M., Kacenelenbogen, M. S., Flynn, C. J., Pistone, K., Knox, N. M., Piketh, S. J., Haywood, J. M., Formenti, P., Mallet, M., Stier, P., Ackerman, A. S., Bauer, S. E., Fridlind, A. M., Carmichael, G. R., Saide, P. E., Ferrada, G. A., Howell, S. G., Freitag, S., Cairns, B., Holben, B. N., Knobelspiesse, K. D., Tanelli, S., L'Ecuyer, T. S., Dzambo, A. M., Sy, O. O., McFarquhar, G. M., Poellot, M. R., Gupta, S., O'Brien, J. R., Nenes, A., Kacarab, M., Wong, J. P. S., Small-Griswold, J. D., Thornhill, K. L., Noone, D., Podolske, J. R., Schmidt, K. S., Pilewskie, P., Chen, H., Cochrane, S. P., Sedlacek, A. J., Lang, T. J., Stith, E., Segal-Rozenhaimer, M., Ferrare, R. A., Burton, S. P., Hostetler, C. A., Diner, D. J., Seidel, F. C., Platnick, S. E., Myers, J. S., Meyer, K. G., Spangenberg, D. A., Maring, H., and Gao, L.: An overview of the ORACLES (ObseRvations of Aerosols above CLouds and their intEractionS) project: aerosol–cloud–radiation interactions in the southeast Atlantic basin, Atmos. Chem. Phys., 21, 1507–1563, https://doi.org/10.5194/acp-21-1507-2021, 2021.